# Splicing factors Sf3A2 and Prp31 have direct roles in mitotic chromosome segregation

Claudia Pellacani[1†], Elisabetta Bucciarelli[1†], Fioranna Renda[2‡], Daniel Hayward[3], Antonella Palena[1], Jack Chen[3], Silvia Bonaccorsi[2], James G Wakefield[3], Maurizio Gatti[1,2]*, Maria Patrizia Somma[1]*

[1]Istituto di Biologia e Patologia Molecolari del CNR, Sapienza Università di Roma, Roma, Italy; [2]Dipartimento di Biologia e Biotecnologie "C. Darwin", Sapienza Università di Roma, Roma, Italy; [3]Biosciences/Living Systems Institute, College of Life and Environmental Sciences, University of Exeter, Exeter, United Kingdom

**\*For correspondence:**
maurizio.gatti@uniroma1.it (MG);
patrizia.somma@uniroma1.it (MPS)

[†]These authors contributed equally to this work

**Present address:** [‡]New York State Department of Health, Wadsworth Center, New York, United States

**Competing interests:** The authors declare that no competing interests exist.

**Abstract** Several studies have shown that RNAi-mediated depletion of splicing factors (SFs) results in mitotic abnormalities. However, it is currently unclear whether these abnormalities reflect defective splicing of specific pre-mRNAs or a direct role of the SFs in mitosis. Here, we show that two highly conserved SFs, Sf3A2 and Prp31, are required for chromosome segregation in both *Drosophila* and human cells. Injections of anti-Sf3A2 and anti-Prp31 antibodies into *Drosophila* embryos disrupt mitotic division within 1 min, arguing strongly against a splicing-related mitotic function of these factors. We demonstrate that both SFs bind spindle microtubules (MTs) and the Ndc80 complex, which in Sf3A2- and Prp31-depleted cells is not tightly associated with the kinetochores; in HeLa cells the Ndc80/HEC1-SF interaction is restricted to the M phase. These results indicate that Sf3A2 and Prp31 directly regulate interactions among kinetochores, spindle microtubules and the Ndc80 complex in both *Drosophila* and human cells.
DOI: https://doi.org/10.7554/eLife.40325.001

## Introduction

Several genome-wide screens carried out both in *Drosophila* and human cells have shown that RNAi-mediated depletion of many different splicing factors (SFs) results in a variety of mitotic defects, ranging from aberrant spindle structure, abnormal chromosome segregation and failure in cytokinesis (*Goshima et al., 2007*; *Kittler et al., 2004*; *Neumann et al., 2010*; *Somma et al., 2008*). Although many studies attributed the observed mitotic phenotypes to defective splicing of specific pre-mRNAs required for cell division (*Burns et al., 2002*; *Maslon et al., 2014*; *Pacheco et al., 2006*; *Sundaramoorthy et al., 2014*; *van der Lelij et al., 2014*), other studies pointed to a direct role of the SFs in mitotic division after the breakdown of the nuclear envelope ('open mitosis') (*Hofmann et al., 2013*; *Hofmann et al., 2010*; *Montembault et al., 2007*).

An example of a splicing defect leading to an aberrant mitotic phenotype is provided by the analysis of mutations in the *S. cerevisiae CEF1* gene, which encodes a conserved SF. In *CEF1* mutants, the failure to remove the single intron of the α-tubulin gene results in reduction of the tubulin level, disrupting mitotic spindle assembly. However, cells containing an engineered intronless α-tubulin gene were resistant to mutations in *CEF1*, indicating that the splicing defect is responsible for the phenotype (*Burns et al., 2002*). Similarly, RNAi-mediated depletion of SFs such as SNW1, PRPF8, MFAP1, NHP2L1, SART1 and CDC5L causes loss of sister chromatid cohesion and correlates with defective pre-mRNA splicing of sororin, a factor required for association of cohesin with DNA. Here again, the expression of an intronless version of sororin partially rescued the mitotic phenotype

elicited by loss of the SFs (*Sundaramoorthy et al., 2014*; *van der Lelij et al., 2014*). An indirect role of SFs in mitotic division is also suggested by studies on the hU2F35 and SRSF1 factors (*Maslon et al., 2014*; *Pacheco et al., 2006*).

However, there are also studies that point to a direct mitotic role of SFs in open mitosis. Depletion of the Prp19 splicing complex from *Xenopus* egg extracts results in defective spindle assembly and impaired microtubule-kinetochore interaction. Because in this system neither transcription nor translation of any message, except Cyclin B, is required for spindle assembly, it has been suggested that Prp19 plays a role in spindle formation that is independent of mRNA splicing (*Hofmann et al., 2013*). Another protein involved in mRNA splicing with a direct mitotic role is PRP4, a kinetochore-associated kinase that mediates recruitment of spindle checkpoint (SAC) proteins at kinetochores (*Montembault et al., 2007*). In addition, it has been recently shown that *Xenopus* SFs interact with kinetochore-associated non-coding RNAs, and are required for recruitment of Cenp-C and Ndc80 at kinetochores (*Grenfell et al., 2017*; *Grenfell et al., 2016*). Therefore, it appears that some SFs are required for the splicing of specific mitotic pre-mRNAs, while others directly participate in the mitotic process. The latter possibility is consistent with the fact that transcription and splicing are suppressed during cell division, allowing SFs to perform direct mitotic functions (*Hofmann et al., 2010*).

Here, we report that the Sf3A2 and Prp31 SFs play direct mitotic functions in both *Drosophila* and human open mitosis. We show that depletion of these SFs affects spindle formation and disrupts chromosome segregation. We also show that antibody-based inhibition of *Sf3A2* or *Prp31* in fly embryos results in a strong and highly specific mitotic phenotype, which manifests less than 1 min following the injection, arguing against an indirect mitotic role of these SFs. Consistent with these results, Sf3A2 and Prp31 bind microtubules (MTs) and the Ndc80 complex that mediates kinetochore-MT attachment. Collectively, our results indicate that Sf3A2 and Prp31 regulate interactions among kinetochores, spindle MTs and the Ndc80 complex.

## Results

### Sf3A2 and Prp31 are required for mitotic chromosome segregation *in Drosophila*

We have previously observed that RNAi against the *Drosophila* homologues of *SF3A2* (*CG10754*; henceforth designated *Sf3A2*) and *PRPF31* (*CG6876*; henceforth *Prp31*) results in abnormal chromosome congression and segregation (*Somma et al., 2008*). Sf3A2 and Prp31 are both found in the *Drosophila* spliceosomal B complex and interact with the U2 and U4/U6 snRNPs, respectively (*Herold et al., 2009*). We began this investigation with a detailed, quantitative cytological examination of mitosis following RNAi against *Sf3A2* or *Prp31* in S2 cells; for these experiments we used dsRNAs targeting the coding regions of these genes (see Materials and methods). To check for RNAi efficiency by western blotting, we raised and affinity-purified two specific antibodies against Sf3A2 and Prp31; western blotting of cell extracts demonstrated that these antibodies specifically recognize bands of the expected molecular weights (33 and 65 kDa, respectively), and that these bands are strongly reduced after RNAi against the corresponding genes (*Figure 1A*, *Figure 1—figure supplement 1*). The cytological consequences of Sf3A2 or Prp31 depletion were examined only in cell populations where the pertinent band was reduced to at least 20% of control level.

*Sf3A2* and *Prp31* RNAi cells showed similar mitotic phenotypes; they exhibited significant increases in both monopolar spindles and PMLES as compared to mock controls. PMLES are peculiar *pro*metaphase-*l*ike cells with *el*ongated *s*pindles associated with chromosomes comprising both sister chromatids (*Figure 1B and C*, *Figure 1—figure supplement 2*). Intriguingly, some PMLES showed central spindle-like structures and irregular cytokinetic rings, while still possessing elevated Cyclin B levels (*Figure 1B*, *Figure 1—figure supplement 2*); they also showed kinetochore-associated BubR1, a SAC protein that normally dissociates from kinetochores at the metaphase-anaphase transition (*Figure 1C*). PMLES are therefore metabolically in a pre-anaphase stage but are nevertheless permissive of typical telophase events, such as central spindle assembly and initiation of cytokinesis. *Sf3A2* and *Prp31* RNAi cells showed a higher frequency of pre-anaphase mitotic figures (prometaphases, metaphases and PMLES) and a lower frequency of ana-telophases compared to controls, suggesting that they suffer of a substantial delay in the metaphase-to-anaphase transition.

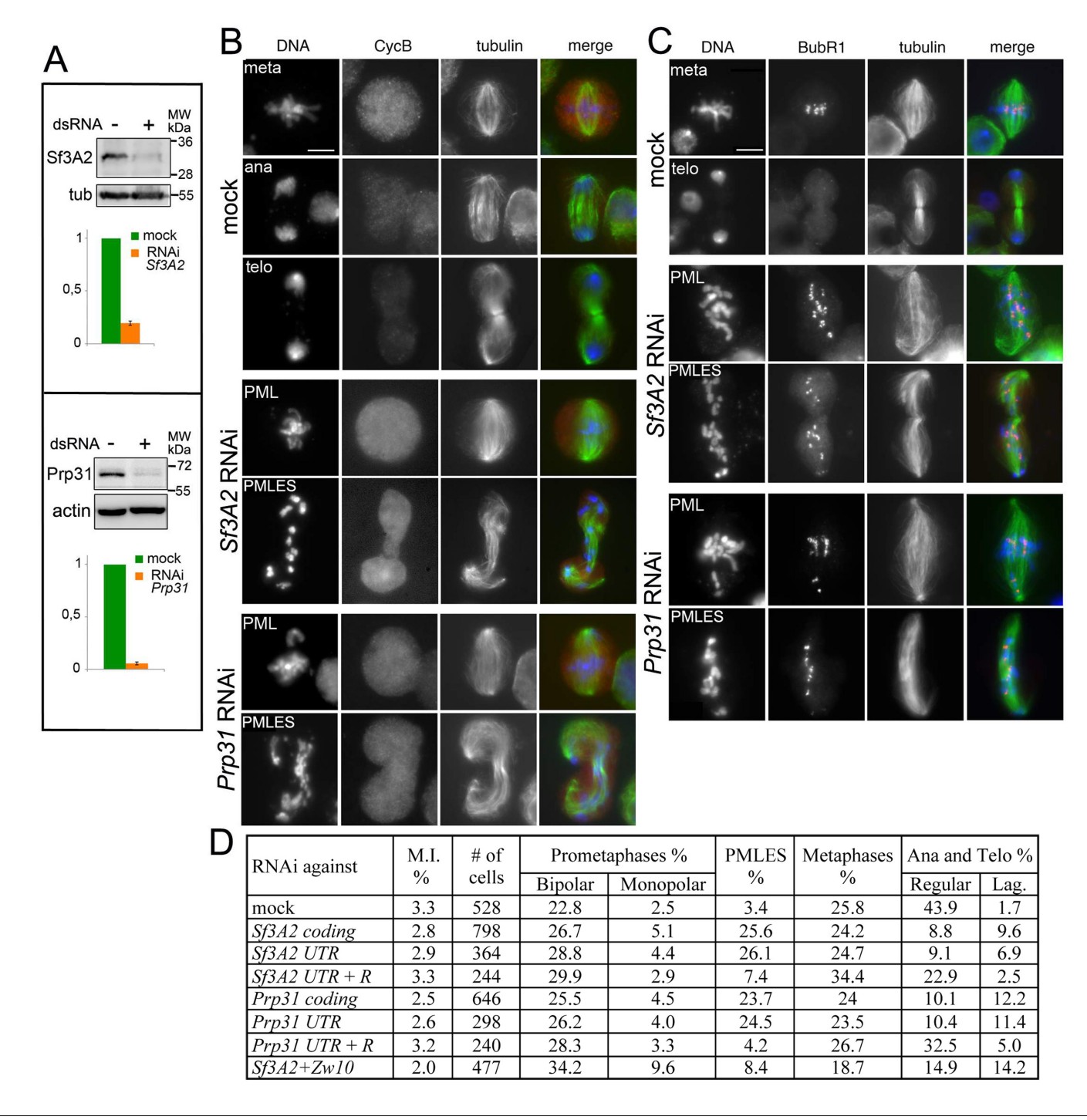

**Figure 1.** RNAi-mediated depletion of Sf3A2 or Prp31 inhibits sister chromatid separation during mitosis. (**A**) Western blots of S2 cell extracts and quantitation of relative band intensities showing that RNAi against *Sf3A2* or *Prp31* strongly reduces the levels of these proteins; tubulin (tub) and actin are loading controls (the full blot is shown in (**Figure 1—figure supplement 1**). (**B**) Mitotic figures observed in mock-treated control cells and in Sf3A2- and Prp31-depleted cells stained for DNA (blue), tubulin (green) and Cyclin B (red). meta, metaphase; ana, anaphase; telo, telophase; PML, prometaphase-like; PMLES *prometaphase-like* cells with *elongated spindles*. Note that PMLES exhibit a high level of Cyclin B. Scale bar, 5 μm. (**C**) Mitotic figures observed in control and RNAi cells stained for DNA (blue), tubulin (green) and BubR1 (red). BubR1 is enriched at the kinetochores in PMLES but not in control ana-telophases. Scale bar, 5 μm. (**D**) Frequencies of mitotic figures observed after RNAi against the indicated *Sf3A2* and *Prp31* sequences, or against both the *Sf3A2* and *Zw10* coding sequences; R are rescue constructs expressing either the *Sf3A2* or the *Prp31* coding

*Figure 1 continued on next page*

*Figure 1 continued*

sequence devoid of the UTRs. Lag, lagging chromosomes between the ana-telophase nuclei. M.I., mitotic index (i.e. percent of cells undergoing mitosis).

DOI: https://doi.org/10.7554/eLife.40325.002

The following source data and figure supplements are available for figure 1:

**Source data 1.** Source data for *Figure 1A* and *Figure 1—figure supplement 4*.

DOI: https://doi.org/10.7554/eLife.40325.007

**Figure supplement 1.** Specificity of the anti-Sf3A2 and anti-Prp31 antibodies.

DOI: https://doi.org/10.7554/eLife.40325.003

**Figure supplement 2.** Examples of monopolar spindles and PMLES observed in *Sf3A2* and *Prp31* RNAi cells.

DOI: https://doi.org/10.7554/eLife.40325.004

**Figure supplement 3.** RNAi against the coding sequences and the UTRs of Sf3A2 and Prp31 rules out off-target effects (see also *Figure 1D*).

DOI: https://doi.org/10.7554/eLife.40325.005

**Figure supplement 4.** RNAi-mediated depletion of Sf3A2 or Prp31 affects spindle structure of larval brain cells and inhibits chromosome segregation.

DOI: https://doi.org/10.7554/eLife.40325.006

We have previously observed frequent PMLES (formerly called *p*seudo-*a*na-*t*elophases, abbreviated as PATs; *Somma et al., 2008*) in S2 cells depleted of the centromere-specific histone H3 Cid/Cenp-A or the kinetochore components Ndc80, Nuf2 and Kmn1/Nsl1 (*Renda et al., 2017*; *Somma et al., 2008*), suggesting that PMLES are a consequence of a defective kinetochore-MT interaction in S2 cells. dsRNAs targeting the 5' UTR of *Sf3A2* or the 3' UTR of *Prp31* produced the same mitotic phenotypes as the dsRNAs directed to the coding sequences; these phenotypes were rescued by expression of RNAi resistant genes (*Sf3A2-RFP* and *Prp31-GFP*) lacking the UTRs (*Figure 1D*, *Figure 1—figure supplement 3*), ruling out the possibility of RNAi off-target effects.

We next performed RNAi against *Sf3A2* or *Prp31* in S2 cells expressing histone-GFP and tubulin-mCherry (*Goshima et al., 2007*) and examined mitosis in living cells. We imaged 14 mitotic divisions of mock-treated control cells. All of them entered anaphase within 45 min after prometaphase/metaphase (P/M), namely, the time when the chromosomes are already fully condensed and begin to align to form the metaphase plate (*Figure 2*, *Video 1*). In contrast, of 35 Sf3A2-depleted dividing cells analyzed, only two entered anaphase within 45 min after P/M, two more displayed a delayed anaphase initiation (54 min and 1 hr 22 min after P/M) accompanied by defective chromosome segregation, while the other 31 cells examined remained in a metaphase-like stage with unseparated sister chromatids for the entire time of filming, which ranged from 1 hr 30 min to more than 4 hr. In about one third of these cells, the chromosomes remained at the center of the cell; in the other two thirds, the chromosomes exhibited oscillations, dispersing along the spindle and within the cytoplasm (*Figure 2*, *Video 2*); these latter cells would appear as PMLES in fixed material. The phenotype of Prp31-depleted cells was similar to that seen in *Sf3A2* RNAi cells. We filmed 12 cells; three entered anaphase after 40, 50 or 85 min after P/M; the other 9 cells examined remained in a metaphase-like stage for the entire time of filming, which ranged from 1 hr 50 min to 3 hr 50 min (*Figure 2*, *Video 3*).

We also asked whether *Sf3A2* and *Prp31* are required for mitotic division in larval brains. We generated flies carrying a suitable *UAS-RNAi* transgene and two constructs bearing the *GAL4* and *GAL80^{ts}* genes, both under the control of a tubulin promoter. Larvae of this genetic constitution were exposed to 29°C for 72 hr to inactivate *GAL80^{ts}* and restore the GAL4 transcriptional activity; they were then dissected and their brains analyzed for mitotic defects; as controls we examined brains from larvae bearing the RNAi construct alone, grown at 29°C for 72 hr. The mitotic divisions observed in these control brains did not display any defect (*Figure 1—figure supplement 4*). In *Sf3A2*-RNAi brains, the spindles were often irregularly shaped; 25% of the cells were monopolar and the frequency of ana-telophases was lower than in controls (15% *vs* 31%). A similar phenotype was observed in *Prp31* RNAi brains, which displayed a substantial increase in monopolar figures and a decrease in ana-telophases compared to controls (*Figure 1—figure supplement 4*). However, in brains we did not observe clear PMLES figures similar to those seen in Sf3A2- or Prp31-depleted S2 cells, suggesting that brain cells possess a control mechanism that prevents spindle elongation in the absence of chromosome segregation and Cyclin B degradation.

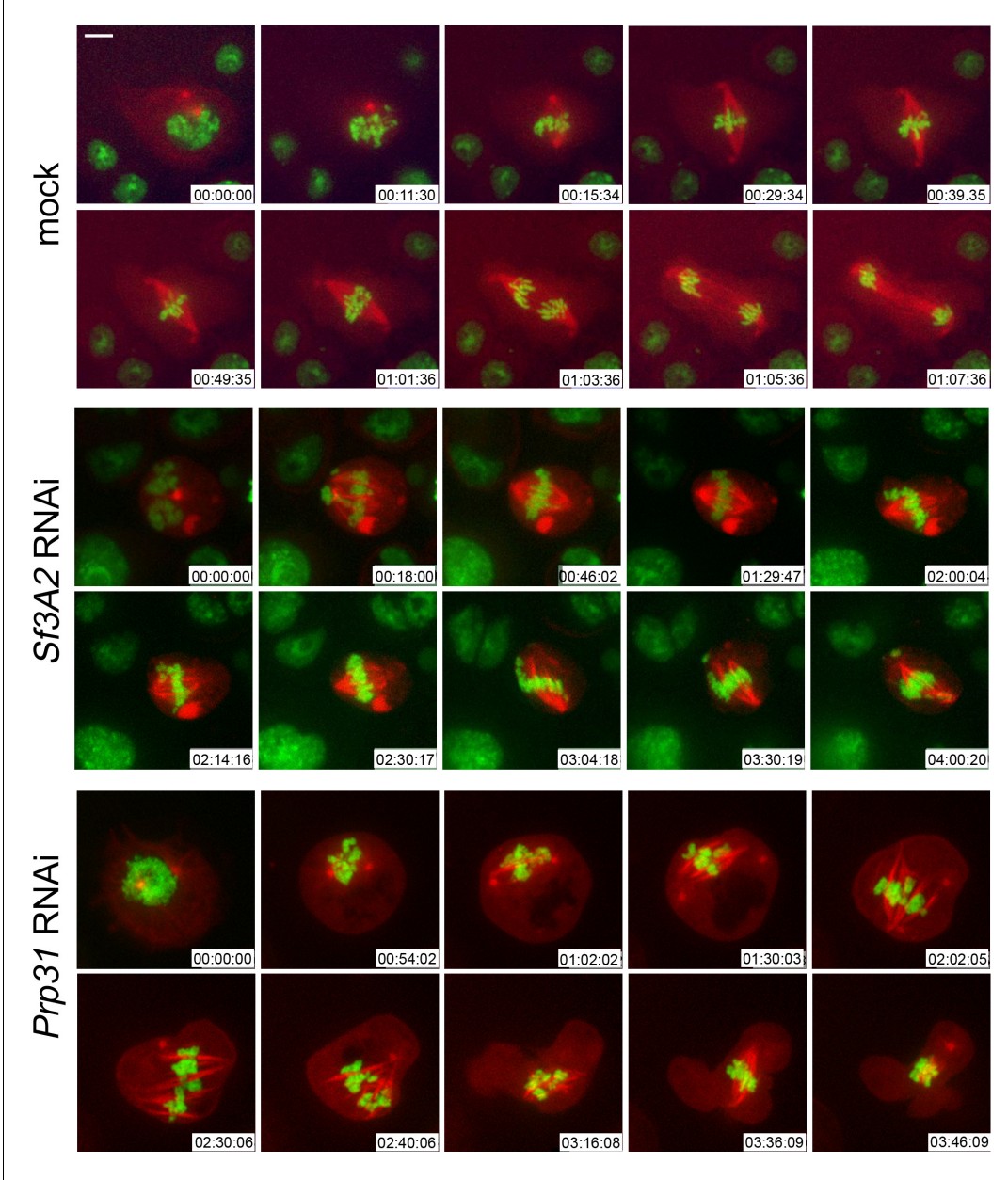

**Figure 2.** Live analysis shows that Sf3A2- and Prp31-depleted cells arrest in metaphase. Stills from time-lapse videos of mitosis in mock-treated and RNAi S2 cells expressing histone-GFP (green) and cherry-tubulin (red). The numbers at the bottom of each frame indicate the time (h:min:s) elapsed from the beginning of imaging. See text for description of chromosome behavior. See also *Videos 1–3*. Scale bar, 5 μm.
DOI: https://doi.org/10.7554/eLife.40325.008

Studies on splicing factors have shown that both Sf3A2 and Prp31 form subcomplexes with specific interacting partners (*Herold et al., 2009*; *Nguyen et al., 2016*). To ask whether the components of these subcomplexes are involved in mitosis, we performed RNAi against *Sf3A1* and *Sf3A3/ noisette* that encode Sf3A2 interactors, and against *Prp3, Prp4/U4-U6-60K, Prp6/CG6841, Prp8* and *hoip/CG3949*, the genes that specify the *Drosophila* homologues of Prp31 interactors. The cytological analysis of these RNAi cells did not reveal obvious mitotic defects (*Figure 3*). However, even if RT PCR could not detect mRNAs from these genes, we cannot completely exclude that some of them encode particularly long-lived proteins that persist after RNAi. Nonetheless, considering that RNAi against *Sf3A2* or *Prp31* strongly reduces the levels of the corresponding proteins, we believe

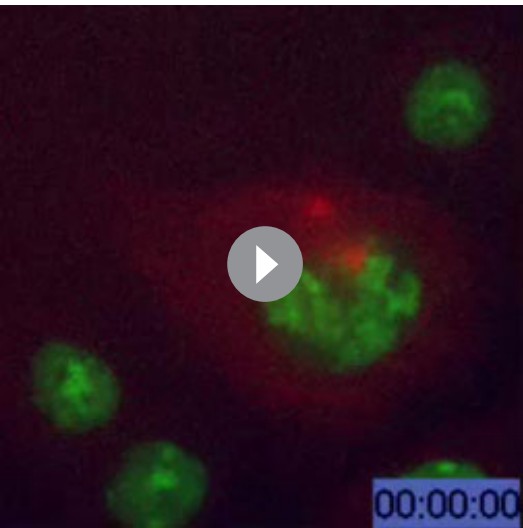

**Video 1.** Mitosis in control S2 cells expressing histone-GFP and tubulin mCherry.
DOI: https://doi.org/10.7554/eLife.40325.009

that at least some of the Sf3A2- and Prp31-interacting factors do not play mitotic roles. This suggests that only individual subunits of the spliceosome subcomplexes, and not the entire subcomplexes, are specifically required for mitosis.

## Depletion of Sf3A2 prevents SAC satisfaction

Cyclin B is normally degraded cell autonomously at the metaphase-to-anaphase transition. However, following RNAi against the SF-coding genes both metaphases and PMLES displayed high Cyclin B levels. In addition, anaphases were infrequent, suggesting that the SAC remains engaged. Further, most prometaphases/metaphases and all PMLES displayed clear accumulations of checkpoint proteins at their kinetochores (BubR1 and Zw10, *Figure 1C* and *Somma et al., 2008*; see also *Figure 7—figure supplement 1* below). To gather evidence that the partial metaphase arrest displayed by SF-depleted cells is due to incapacity to satisfy the SAC we asked whether disruption of the SAC machinery allows these cells to enter anaphase. We thus performed *Sf3A2 zw10* double RNAi. Zw10 is required for Mad2 localization to the kinetochores and efficient SAC (*Buffin et al., 2005*). As shown in *Figure 1D*, *Sf3A2 zw10* double RNAi cells displayed a three-fold decrease in the frequency of PMLES and an almost two-fold increase in the frequency of ana-telophases compared to cells depleted of Sf3A2 only. These results indicate that in Sf3A2-depleted cells the SAC is not satisfied and this is at least in part responsible for the observed phenotype.

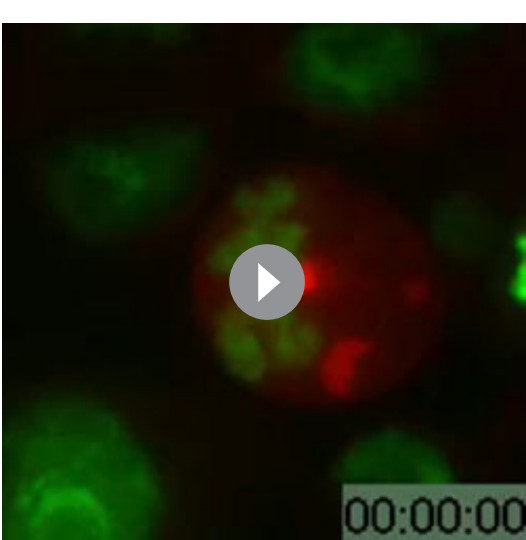

**Video 2.** Prolonged metaphase arrest in Sf3A2-depleted S2 cells expressing histone-GFP and tubulin mCherry.
DOI: https://doi.org/10.7554/eLife.40325.010

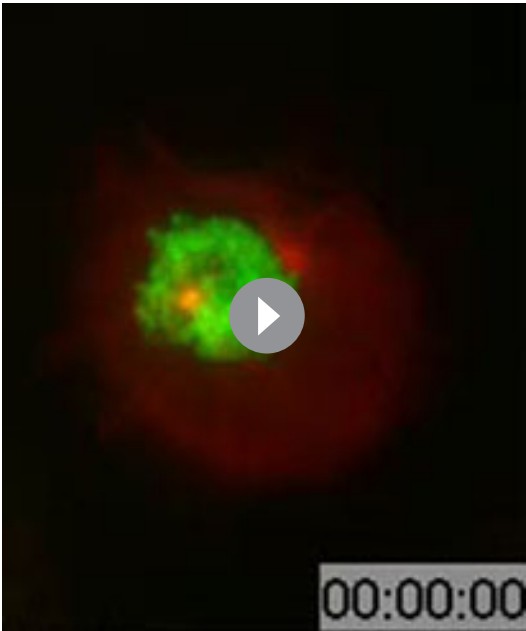

**Video 3.** Prolonged metaphase arrest in Prp31-depleted S2 cells expressing histone-GFP and tubulin mCherry.
DOI: https://doi.org/10.7554/eLife.40325.011

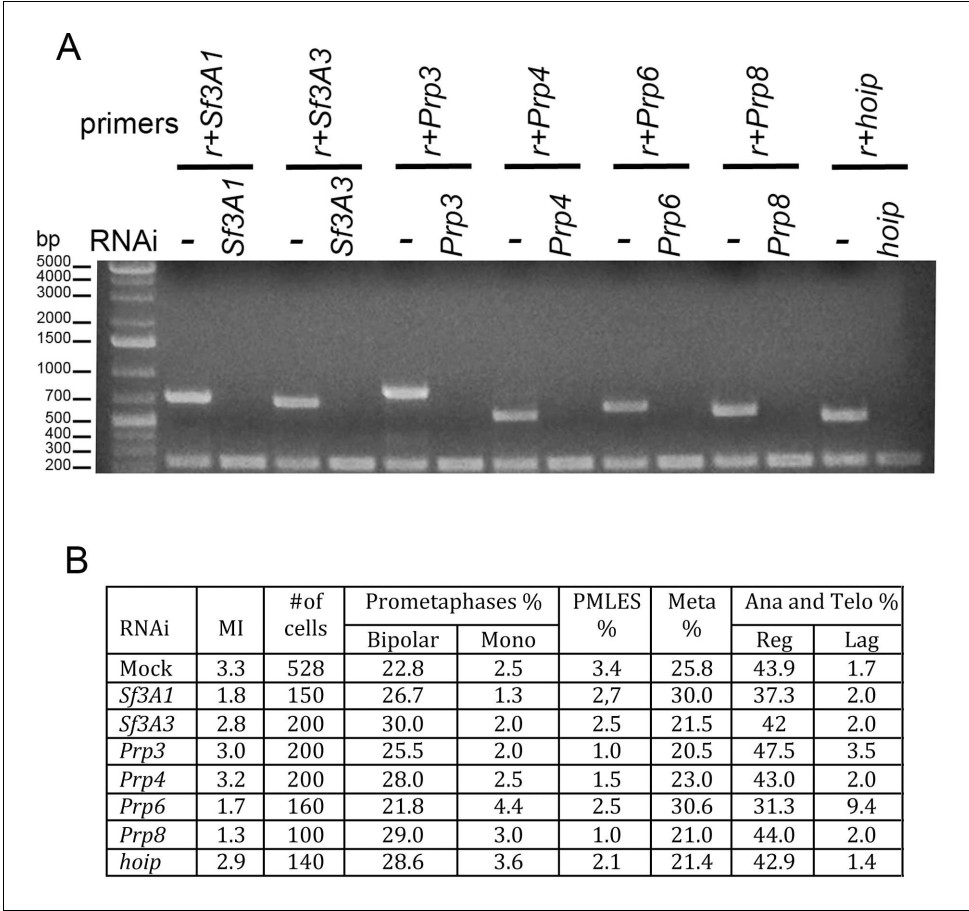

**Figure 3.** RNAi against *SfA31*, *Sf3A3*, *Prp3*, *Prp4*, *Prp6*, *Prp8*, or *hoip/CG3949* does not result in gross mitotic defects. (**A**) RT PCR showing that RNAi against the indicated factor strongly reduces the corresponding transcript; r, *rp49* used as internal control. (**B**) Mitotic indexes (MI) and frequencies of mitotic figures observed after RNAi against the indicated gene. Mono, monopolar; Meta, metaphases; Ana and Telo, anaphases and telophases; Reg, regular; Lag, lagging chromosomes between the ana-telophase nuclei. The control (Mock) is the same as that of *Figure 1*.

DOI: https://doi.org/10.7554/eLife.40325.012

## The phenotypes elicited by Sf3A2 or Prp31 inhibition in the embryo suggest a direct mitotic role of these SFs

To discriminate between a direct and an mRNA splicing-mediated mitotic role of Sf3A2 and Prp31, we sought to acutely perturb their functions. We therefore injected syncytial *Drosophila* embryos expressing histone-RFP and tubulin-GFP with highly specific affinity-purified antibodies directed to these SFs (see *Figure 1—figure supplement 1C*), with unrelated antibodies or the buffer only. If not otherwise specified, imaging started no more than 30 s after injection. We analyzed mitosis in 10 embryos injected with the anti-Sf3A2 antibody and in 10 embryos injected with the anti-Prp31 antibody. Embryos were injected in (i) interphase; in many cases, shortly before nuclear envelope breakdown (NEB) that occurs in early prometaphase, as suggested by the fact that they were in late prophase/early prometaphase at beginning of filming, (ii) during early prometaphase (showing full chromosome condensation by the beginning of filming) or (iii) during late telophase and then monitored throughout the following cell cycle. An additional five embryos were injected with buffer alone before NEB to serve as mock controls. We carefully compared mitotic progression in these different injection conditions to ascertain the nature of specific phenotypes and their temporal manifestation.

In movies starting just before NEB, the time elapsing from NEB to initial chromosome alignment was similar between control-injected and α-Sf3A2- or α-Prp31-injected embryos. However, the

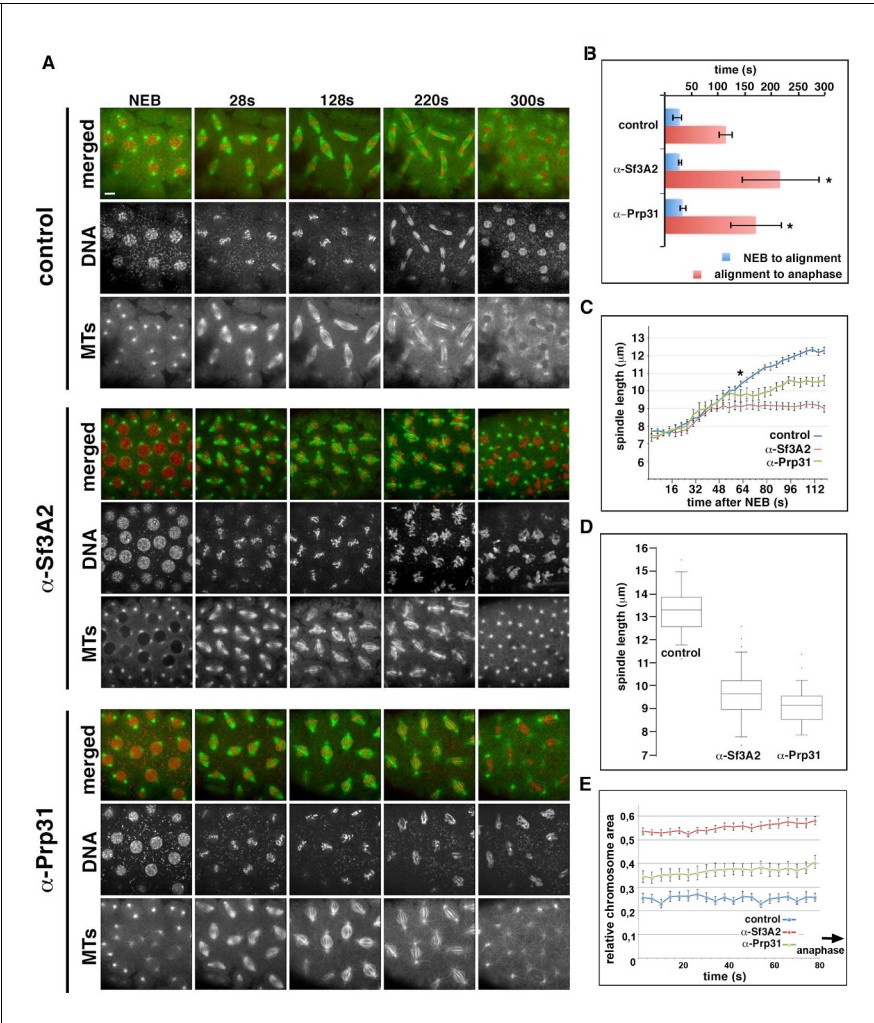

**Figure 4.** Acute inhibition of Sf3A2 and Prp31 causes mitotic delay, spindle instability and abortive chromosome segregation in *Drosophila* embryos. (A) Stills from time-lapse movies of syncytial embryos expressing GFP-tubulin and histone-RFP injected with BSA (control), α-Sf3A2 or α-Prp31 antibodies. Scale bar, 5 μm. (B) Bar chart comparing mitotic timing between control injected and α-Sf3A2 or α-Prp31 injected embryos. The time from NEB to initial chromosome alignment is similar between conditions but the length of metaphase is significantly increased in both α-Sf3A2 and α-Prp31 injected embryos (*p=0.016 control: α-Sf3A2; p=0.029 control: α-Prp31; unpaired t-test, five embryos each condition); error bars, SD. (C) Spindle length over time in the control, α-Sf3A2 and α-Prp31 injected embryos. From 64 s following NEB onwards, the shorter length of spindles in anti-SF injected embryos becomes significant (*; unpaired t-test; eight spindles per embryo). Error bars, SEM. (D) Box and whisker plots showing the quartile ranges of mitotic spindle length ~20 s prior to anaphase onset. Error bars, SD. (E) Area occupied by the chromosomes during the last 80 s of filming before anaphase onset, relative to the area occupied in early prophase. The areas were measured using the ImageJ software; 15 metaphases analyzed for each condition; p<0.001 control: α-Sf3A2; p<0.001 control: α-Prp31; unpaired t-test). Error bars, SEM. See also *Videos 4–6*.
DOI: https://doi.org/10.7554/eLife.40325.013

The following source data and figure supplements are available for figure 4:

**Source data 1.** Source data *Figure 4B–E* and *Figure 4—figure supplement 2D and E*.
DOI: https://doi.org/10.7554/eLife.40325.017

**Figure supplement 1.** Mitotic division in live *Drosophila* embryos injected with anti-Sf3A2 or anti-Prp31 antibodies during the late telophase of the previous cell cycle.
DOI: https://doi.org/10.7554/eLife.40325.014

**Figure supplement 2.** Failure of mitotic chromosome segregation in live *Drosophila* embryos injected with anti-Sf3A2 or anti-Prp31 antibodies during mitosis.
DOI: https://doi.org/10.7554/eLife.40325.015

**Figure supplement 3.** Dynamic localization of fluorescently-labeled anti-SF3A2 and anti-Prp31 antibodies during *Drosophila* syncytial divisions.
DOI: https://doi.org/10.7554/eLife.40325.016

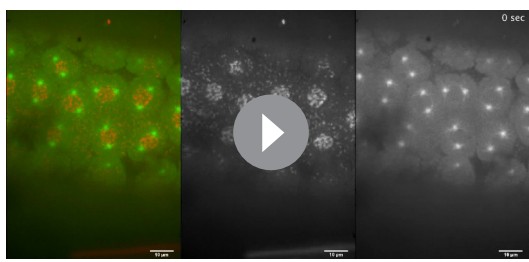

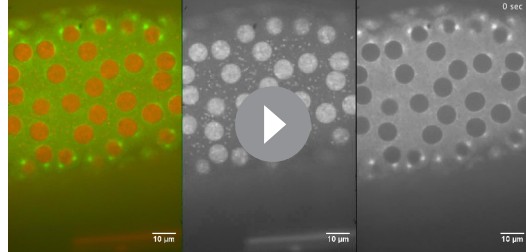

**Video 4.** Mitotic division in mock-injected (control) embryos expressing histone RFP and tubulin-GFP; imaging begins before NEB.
DOI: https://doi.org/10.7554/eLife.40325.018

**Video 5.** Mitotic division in α-Sf3A2-injected embryos expressing histone RFP and tubulin-GFP; imaging begins before NEB.
DOI: https://doi.org/10.7554/eLife.40325.019

length of metaphase was significantly increased in embryos treated with the anti-SF antibodies (*Figure 4A and B*, *Videos 4–6*). In addition, in both α-Sf3A2- and α-Prp31-injected embryos, spindles did not increase to mature metaphase spindle length, as occurred with controls, but remained 'stunted'. This decrease in spindle length relative to controls became significant at 64 s following NEB and remained significant until anaphase (*Figure 4A and C*, *Videos 5* and *6*), such that, 20 s prior to anaphase onset, the spindles of anti-SF injected embryos were approximately 30% shorter than those of controls (*Figure 4A and D*). Our qualitative observations also suggested that the chromosomes of α-Sf3A2- and α-Prp31-injected embryos were less aligned than in controls throughout the extended metaphase period (*Figure 4A*, *Videos 5* and *6*). To test this, we measured the area encompassed by the chromosomes during the last 80 s (20 frames) that precede anaphase in anti-SF injected embryos. The chromosomes were indeed less tightly aligned than in their mock-injected counterparts (*Figure 4E*). Finally, the most striking defect in anti-SF injected embryos was a complete failure in chromosome segregation during anaphase, such that chromosomes decondensed and coalesced in a single indistinct mass (*Figure 4A*, *Videos 5* and *6*). *Figure 4A* and *Video 6* show a specific example of an embryo injected with anti-Prp31antibodies. In this case, NEB occurs 16 s after the movie begins. As it takes 15–30 s to begin filming following injection, we can unequivocally conclude that the spindle length phenotype manifests within 110 s (30 + 16 + 64) following injection.

To more robustly distinguish between a direct or indirect role in mitosis for the SFs, we next analyzed movies of embryos injected at different stages of the cell cycle. Embryos injected with either of the SF antibodies during late telophase did not exhibit an increase in the length of the following S phase compared to the mock-injected control nuclei, suggesting interphase/S phase was not disrupted. However, when these embryos entered mitosis they displayed the same defects seen in embryos injected just before or during mitosis (*Figure 4—figure supplement 1*). Further, embryos which were in metaphase immediately following injection with α-Sf3A2 or α-Prp31 antibodies manifested the same phenotype as those injected during interphase – with short spindles, poorly aligned chromosomes and failure in segregation (*Figure 4—figure supplement 2*, *Videos 7* and *8*). As no more than 30 s elapsed between injection and the start of imaging, and the short spindle phenotype was clear since the beginning of filming (*Figure 4—figure supplement 2*), we conclude that the mitotic phenotype observed upon anti-SF injection occurs in less than 1 min. This is inconsistent with the time required for splicing and translation and strongly supports a direct role for the SFs in mitosis.

Finally, to rule out the possibility that the specific phenotypes observed after anti-SF injection were a spurious non-specific result, we injected embryos with an antibody against Dgt6 (a subunit

**Video 6.** Mitotic division in α-Prp31-injected embryos expressing histone RFP and tubulin-GFP; imaging begins before NEB.
DOI: https://doi.org/10.7554/eLife.40325.020

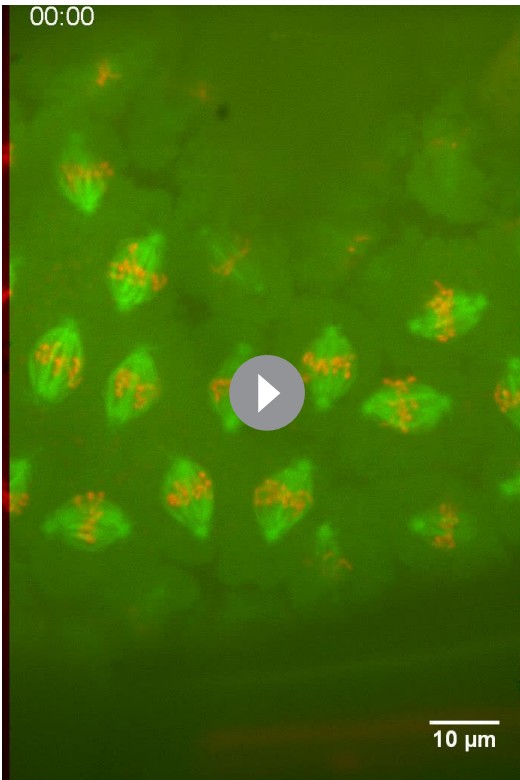

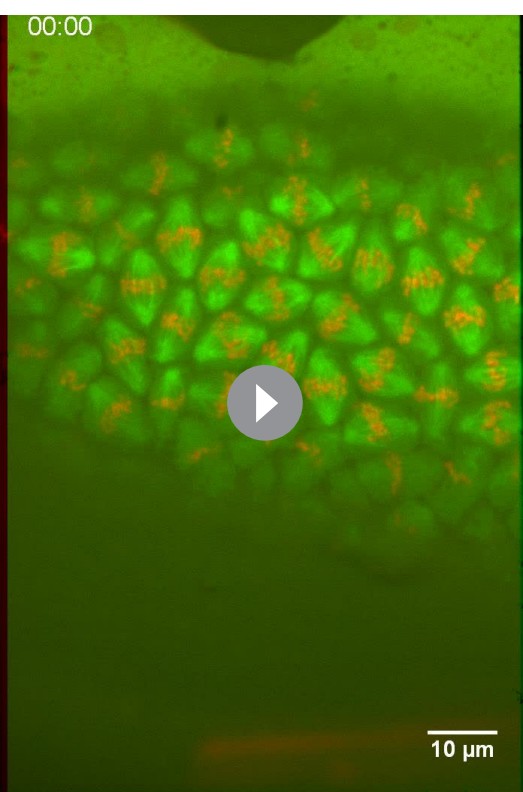

**Video 7.** Mitotic division in α-Sf3A2-injected embryos expressing histone RFP and tubulin-GFP; imaging begins after NEB.
DOI: https://doi.org/10.7554/eLife.40325.021

**Video 8.** Mitotic division in α-Prp31-injected embryos expressing histone RFP and tubulin-GFP; imaging begins after NEB.
DOI: https://doi.org/10.7554/eLife.40325.022

of the Augmin complex involved in spindle formation) that had been prepared at the same time as the anti-SF antibodies. Consistent with previous results (*Hayward et al., 2014*), the embryos injected with the anti-Dgt6 antibodies showed delayed spindle formation, thinner and long spindles and arrested in a metaphase-like state. This phenotype is very different from that elicited by anti SF-antibodies, ruling out the possibility that the mitotic effects of the anti-SF antibodies are due to chemical contaminants associated with antibody purification and concentration processes.

To supplement these phenotypic observations, we determined the localization of Sf3A2 and Prp31 in living embryos. In *Drosophila* embryos, mitosis is semi-open; the spindle is established through polar fenestrae formed during early prometaphase and remains surrounded by a nuclear envelope till anaphase (*Harel et al., 1989*; *Kiseleva et al., 2001*). We injected fluorescently labeled α-Sf3A2 or α-Prp31 antibodies into live syncytial embryos and filmed their behavior (*Figure 4—figure supplement 3*). Both antibodies were virtually excluded from interphase and prophase nuclei. However, as soon as the nuclear envelope became fenestrated and the spindle was formed, the antibodies quickly penetrated into the nuclear space and remained there until anaphase; they then became dispersed in the cytoplasm and failed to concentrate in telophase nuclei (*Figure 4—figure supplement 3*). These results strongly suggest that during interphase the antibodies cannot disrupt the splicing process that occurs within the nucleus. Additionally, even if the antibodies are injected one minute before nuclear fenestration, they can act on kinetochores/microtubules only after their entry into the nucleus at prometaphase. Thus, given that splicing does not occur during mitosis (*Hofmann et al., 2010*; *Shin and Manley, 2002*), our embryo injection experiments make highly unlikely an indirect, splicing-dependent mitotic effect of Sf3A3 and Prp31 inhibition.

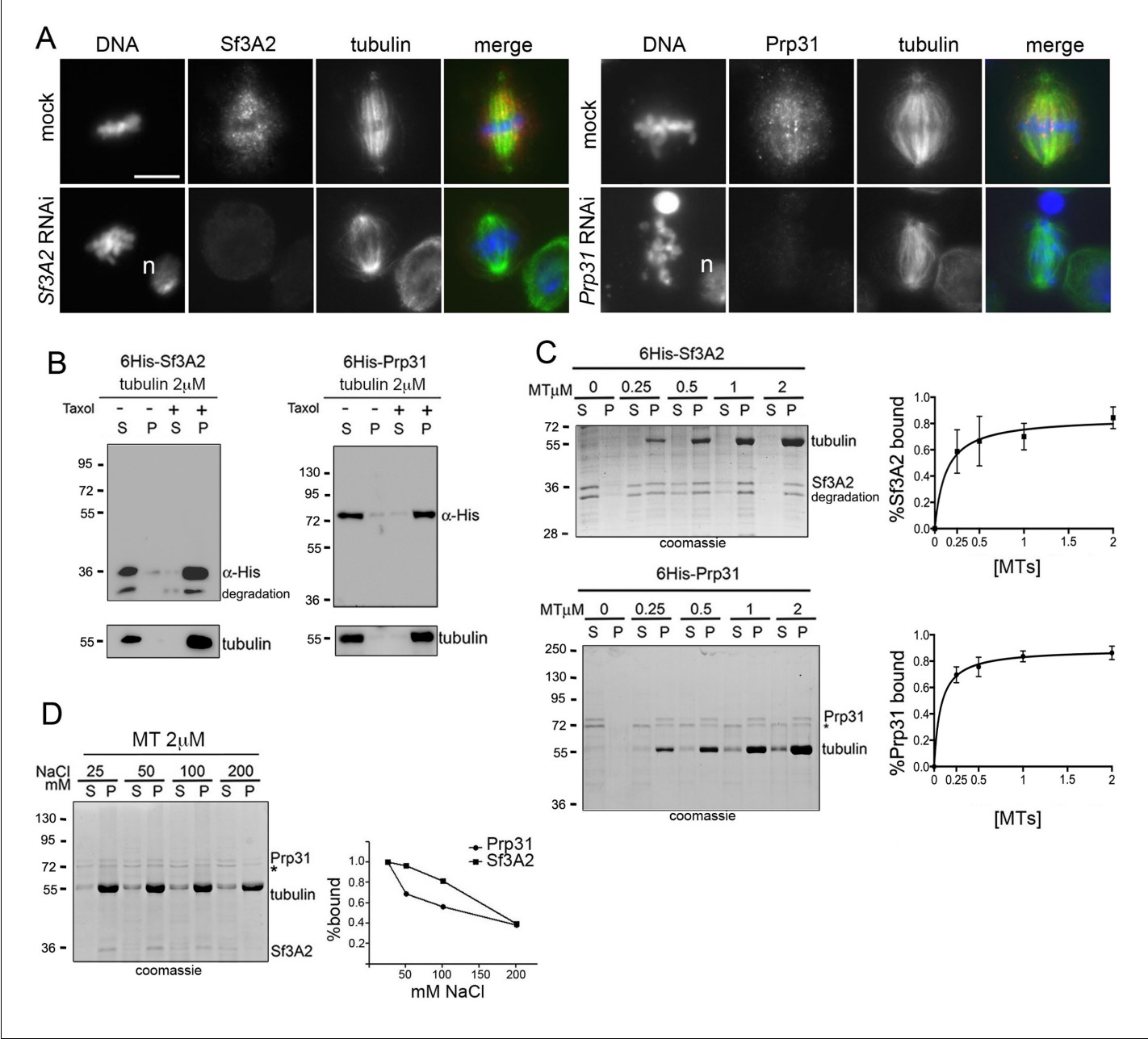

**Figure 5.** Sf3A2 and Prp31 associate with S2 cell spindle MTs. (**A**) Mitotic figures observed in mock-treated control cells and in Sf3A2- and Prp31-depleted cells stained for DNA (blue), tubulin (green) and Sf3A2 or Prp31 (red). Sf3A2 and Prp31 associate with spindle MTs in S2 control cells; note that in *Sf3A2* and *Prp31* RNAi cells spindles are not stained. n, nucleus. Scale bar, 5 μm. (**B**) 6His-Sf3A2 and 6His-Prp31 co-sediment with 2 μM taxol-stabilized MTs (2 μM is the concentration of tubulin dimers in MT polymers) but remain in the supernatant in the presence of unpolymerized tubulin. S, supernatant; P, pellet. Samples were subjected to Western blotting. Tubulin is shown as a control. (**C**) Cosedimentation of 6His-Sf3A2 and 6His-Prp31 with taxol-stabilized MTs at the indicated concentrations in the presence of 25 mM NaCl. The input in the Sf3A2 experiment shows an additional band that is roughly 2 kDa smaller than full-length, 6X-His tagged Sf3A2. This smaller band appears only inconsistently in various preparations and is likely a degradation product because its behavior in these assays is identical to that of the full-length protein. * Unspecific band. Samples were subjected to SDS-PAGE followed by Coomassie staining. Cosedimentation of each SF with MTs was quantified and plotted using data from three independent experiments. Each point in MT-binding curves represents the percentage of MT-bound 6His-Sf3A2 or 6His-Prp31 (intensity of the pellet band/intensity of the pellet plus supernatant bands). Error bars, SD. (**D**) Cosedimentation of SFs with 2 μM taxol-stabilized MTs in the presence of increasing concentrations of NaCl. * Unspecific band. Samples were subjected to SDS-PAGE followed by Coomassie staining. The SF-MT-binding curves are from two independent experiments.

DOI: https://doi.org/10.7554/eLife.40325.023

*Figure 5 continued on next page*

*Figure 5 continued*

The following source data and figure supplements are available for figure 5:

**Source data 1.** Source data for *Figure 5*.

DOI: https://doi.org/10.7554/eLife.40325.026

**Figure supplement 1.** Localization of Sf3A2 and Prp31 during mitosis of S2 cells.

DOI: https://doi.org/10.7554/eLife.40325.024

**Figure supplement 2.** Localization of Sf3A2 and Prp31 on metaphase spindle is MT-dependent and RNA-independent.

DOI: https://doi.org/10.7554/eLife.40325.025

## Sf3A2 and Prp31 associate with the spindle MTs

To determine the localization of Sf3A2 and Prp31 during mitosis we co-stained S2 cells with anti-tubulin antibodies and antibodies to the individual SFs. During interphase and prophase both Sf3A2 and Prp31 were strongly enriched into the nucleus, as expected for SFs; in prometaphase cells both SFs showed a diffuse distribution (*Figure 5*, *Figure 5—figure supplement 1*). However, in metaphase figures with tightly aligned chromosomes, both Sf3A2 and Prp31 were enriched in the central part of the spindle but excluded from the spindle poles and the asters (*Figure 5A*, *Figure 5—figure supplement 1A,B*). During anaphase, Sf3A2 and Prp31 localized at the center of the dividing cells, while in telophase cells they were not bound to spindle but accumulated into the reforming nuclei. Cells depleted of Sf3A2 or Prp31 did not show any spindle staining after incubation with the pertinent antibody (*Figure 5A*). To obtain additional evidence for localization of the SFs in the spindle we examined cells expressing *Sf3A2-RFP* or *Prp31-GFP* under the control of the actin promoter, and found an enrichment of the tagged proteins in the metaphase spindles (*Figure 5—figure supplement 1C*).

To determine whether the spindle staining was due to an association of the SFs with the MTs we incubated the cells for 30 min in 100 µM colchicine, a treatment that is sufficient to depolymerize the spindle MTs in S2 cells but insufficient to cause diffusion of the spindle matrix components (*Schweizer et al., 2013*). We found that after this treatment both Sf3A2 and Prp31 are diffuse and do not show any enrichment in any spindle-like structure (i. e. the spindle matrix). In contrast, RNAse-treated metaphases displayed a normal spindle staining with anti-Sf3A2 and anti-Prp31 antibodies (*Figure 5—figure supplement 2*). Thus, Sf3A2 and Prp31 associate with spindle MTs during metaphase and this association is not mediated by RNA molecules.

The association of *Drosophila* Sf3A2 and Prp31 with metaphase spindles suggests that these proteins may have the ability to directly bind MTs. To provide direct evidence that *Drosophila* Sf3A2 and Prp31 bind MTs, we performed in vitro MT co-sedimentation assays with purified, His-tagged versions of these SFs. We found that, while neither SF was present in the pellet fraction in the presence of unpolymerized tubulin, over 80% of either 6His-Sf3A2 or 6His-Prp31 co-sediments with 2 µM taxol-stabilized MTs (*Figure 5B*). Incubation with increasing amounts of taxol-stabilized MTs (0.25–2 µM of tubulin dimers in MT polymers) confirmed that polymerized MTs were able to precipitate at least 80% of either 6His-Sf3A2 or 6His-Prp31 (*Figure 5C*). Furthermore, increasing NaCl concentration from 50 to 200 mM strongly reduced MT binding of 6His-Sf3A2 or 6His-Prp31, demonstrating that the MT-SF interaction is based on ionic attraction (*Figure 5D*). These in vitro results, coupled with the finding that the mouse orthologue of Sf3A2 binds MTs (*Takenaka et al., 2004*), provide a biochemical basis to support the conclusion that Sf3A2 and Prp31 associate with the MTs of metaphase spindles.

## Sf3A2 and Prp31 silencing does not affect the global or mitotic proteome

Although our embryo injection data strongly suggest a direct role for Sf3A2 and Prp31 in *Drosophila* mitosis, the possibility exists that the SF RNAi phenotypes observed in S2 cells could be a combination of direct effects on mitosis and indirect effects due to failures in mitotic mRNAs splicing. We reasoned that, if the spindle and chromosome defects observed following SF RNAi were due to lack of splicing, the levels of general or specific mitotic proteins would be reduced, in comparison to control cells. To comprehensively test this, we performed a simultaneous Tandem Mass Tag (TMT) quantitative proteomic analysis of control S2 cells, and of cells depleted of Sf3A2, Prp31 or Sf3A1, a

splicing factor that does not exhibit any apparent mitotic defect upon RNAi (*Figure 3*). This quantitative proteomic analysis identified 4483 protein IDs in control S2 cells (the full dataset can be found at www.thewakefieldlab/ms). The abundance scores for the resultant protein IDs from each of the three RNAi samples (*Sf3A1*, *Sf3A2* and *Prp31*) were normalised against the control sample to create an abundance ratio. The mean abundance ratio and standard deviation (SD) were calculated and proteins with abundance ratios of more than 1 SD from the mean were identified and classed as significantly reduced, in comparison to control. For *Sf3A2* and *Prp31* RNAi, this identified 99 and 136 proteins, respectively. Prp31 was the most reduced protein in *Prp31* RNAi cell extracts, while Sf3A2 was the second most reduced protein in *Sf3A2* RNAi cell extracts, demonstrating the validity of the comparative mass spectrometry (MS) approach (*Supplementary file 1*). Interestingly, *Sf3A1* RNAi resulted in 948 proteins significantly reduced, suggesting knock down of *Prp31* and *Sf3A2* has a less general effect on splicing than Sf3A1.

To identify whether any proteins involved in mitosis that could explain the RNAi phenotype were specifically reduced in the *Sf3A2* and *Prp31* RNAi cell extracts, the significantly reduced protein IDs for each condition were interrogated using a Gene Ontology (GO) classifier (GOTermMapper (https://go.princeton.edu/cgi-bin/GOTermMapper), concentrating on the GO terms 'cell division' (GO:0051301), 'mitotic cell cycle' (GO:0000278) and 'chromosome segregation' (GO:0007059). A total of 9 IDs, 19 IDs and 50 IDs were classified with at least one of these GOs for *Sf3A2*, *Prp31* and *Sf3A1* RNAi, respectively (*Figure 6*, *Supplementary file 1*). *Figure 6* shows a Venn diagram, comparing these IDs for each RNAi condition. Only a single protein, HUS1-like, is significantly reduced in both *Prp31* and *Sf3A2*, but not *Sf3A1*, RNAi cells. As HUS1-like is involved in homologous recombination repair during meiosis (*Peretz et al., 2009*), it is unlikely to be the causative factor in the SF phenotype observed in S2 cells. Thus, we conclude there is no evidence that the mitotic phenotypes seen upon Prp31 or Sf3A2 RNAi in S2 cells are a result of reduced protein levels, caused by defective splicing.

## Sf3A2 and Prp31 bind the Ndc80 complex and are required for its correct localization

The observation that cells depleted of either Sf3A2 or Prp31 exhibit frequent PMLES prompted us to investigate whether they normally recruit centromere and kinetochore components. The *Drosophila* centromere/kinetochore contains many proteins and protein complexes that are conserved from flies to humans, including the core centromeric components Cid (CenpA) and Cenp-C, the kinetochore complexes Mis12 (Mis12, Nnf1a, Nnf1b, Kmn1) and Ndc80 (Ndc80, Nuf2, Spc25/Mitch), the chromosomal passenger protein Aurora B, and the SAC components BubR1 and Zw10 (*Liu et al., 2016*; *Przewloka et al., 2007*; *Schittenhelm et al., 2007*; *Williams et al., 2007*). Indirect immunofluorescence of S2 cells depleted of either SF *via* RNAi, showed that loss of either protein does not affect the localization of Cid, Cenp-C, Mis12, Kmn1, AurB, BubR1, or Zw10 (*Figure 1C*, *Figure 7—figure supplement 1*). However, Sf3A2 or Prp31 depletion strongly reduced kinetochore localization of Ndc80 and Spc25/Mitch (*Figure 7A–C*). Crucially, this effect was not due to a reduction in the Ndc80 protein, as western blotting showed no significant difference in the Ndc80 levels between control and *Sf3A2* or *Prp3* RNAi cell extracts, consistent with our proteomic analyses (*Figure 6*, *Supplementary file 1*),

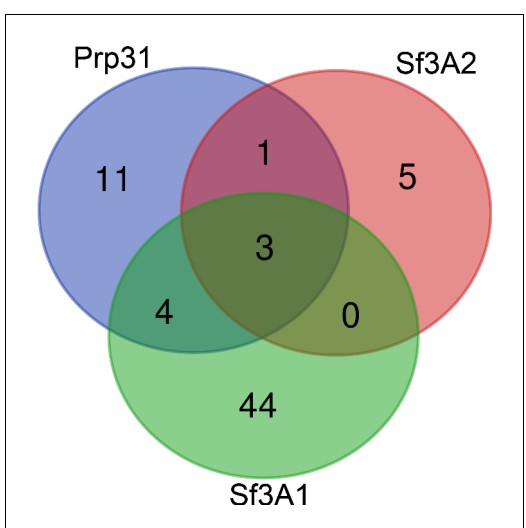

**Figure 6.** Venn diagram showing the overlap of mitotic proteins that are significantly reduced in *Sf3A2*, *Prp31* and *Sf3A1* RNAi cells. Proteins whose abundance was significantly reduced in *Prp31*, *Sf3A2* or *Sf3A1* RNAi cells, and possessing Gene Ontologies (GOs) of mitotic cell cycle, cell division or chromosome segregation, were visualized using the Venn diagram software. Only a single protein, HUS1-like, is shared between Prp31 and Sf3A2, but not Sf3A1.

DOI: https://doi.org/10.7554/eLife.40325.027

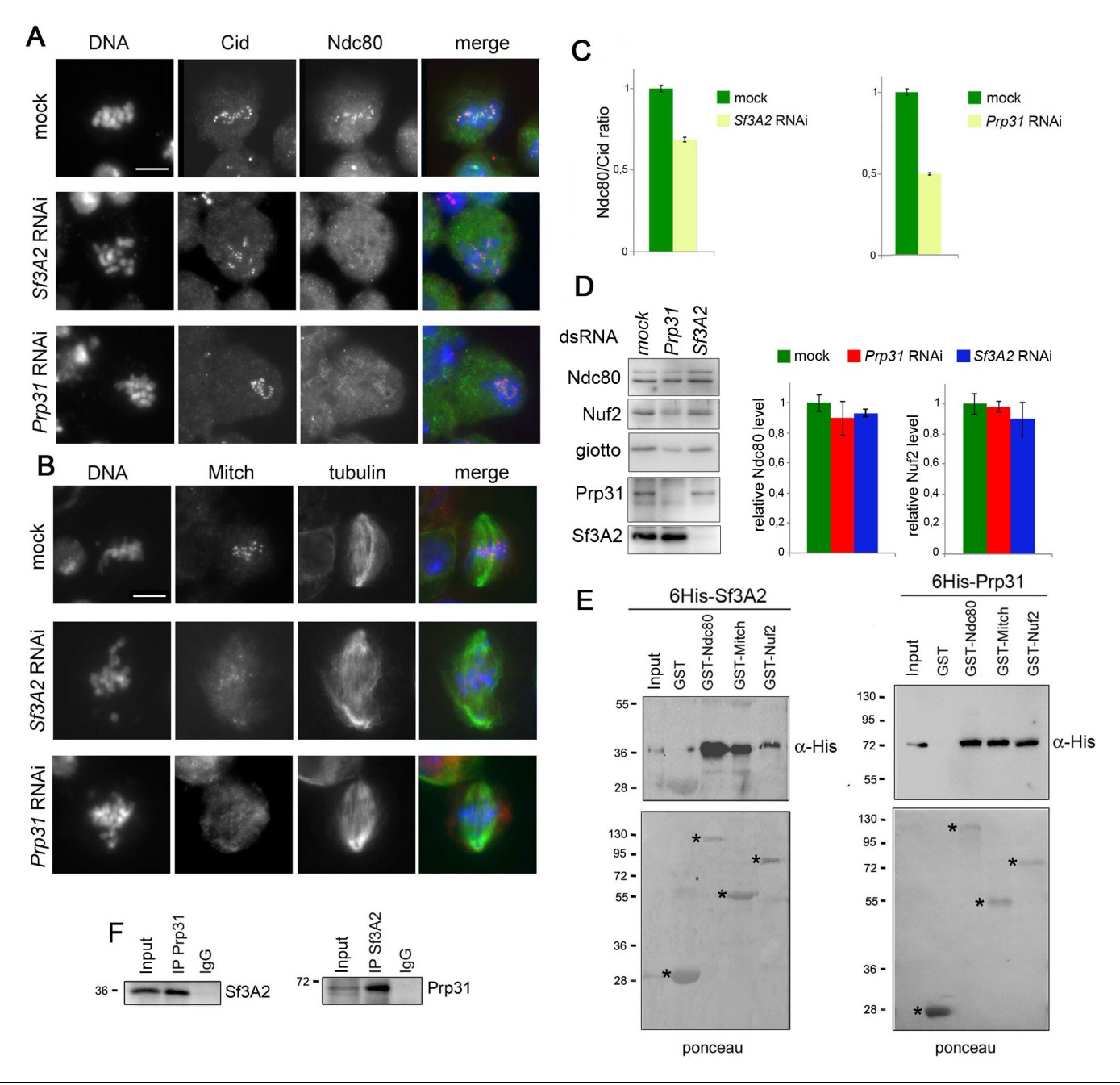

**Figure 7.** Sf3A2 and Prp31 bind Ndc80 and Spc25/Mitch and are required for their localization at kinetochores. (**A**) Metaphase and prometaphase-like figures stained for DNA (blue), Cid/Cenp-A (red) and Ndc80 (green) showing that RNAi against *Sf3A2* or *Prp31* reduces Ndc80 accumulation at kinetochores. Scale bar, 5 µm. (**B**) Metaphase and prometaphase-like figures stained for DNA (blue), Mitch (red) and tubulin (green) showing that RNAi against *Sf3A2* or *Prp31* reduces Spc25/Mitch accumulation at kinetochores. Scale bar, 5 µm. (**C**) Quantification of Ndc80 kinetochore-associated fluorescence relative to the Cid fluorescence in *Sf3A2* and *Prp31* RNAi cells. The relative fluorescence was calculated by measuring the fluorescence (minus the background) of Ndc80/Cid signals of at least 20 cells, using the ImageJ software. Error bars, SEM. (**D**) Western blotting analysis showing that RNAi-mediated depletion of either Sf3A2 or Prp31 does not significantly affect the Ndc80 and Nuf2 amount. Giotto (a *Drosophila* Phosphatidylinositol transfer protein (*Giansanti et al., 2006*)) is used as a loading control. Quantification of the Ndc80 and Nuf2 levels is from three independent experiments. Error bars, SEM. (**E**) GST-Ndc80, GST-Mitch and GST-Nuf2 precipitate bacterially expressed and purified 6His-Sf3A2 (5% input) and 6His-Prp31 (10% input), which are not pulled down by GST alone. Asterisks indicate GST and GST-tagged kinetochore proteins. (**F**) Reciprocal Co-IPs showing that Sf3A2 and Prp31 interact in vivo.

DOI: https://doi.org/10.7554/eLife.40325.029

The following source data and figure supplements are available for figure 7:

*Figure 7 continued on next page*

*Figure 7 continued*

**Source data 1.** Source data for *Figure 7C and D*.
DOI: https://doi.org/10.7554/eLife.40325.033
**Figure supplement 1.** Cenp-C, Mis12, Aurora B, Kmn1 and Zw10 localize normally in *Sf3A2* and *Prp31* RNAi cells.
DOI: https://doi.org/10.7554/eLife.40325.030
**Figure supplement 2.** Colchicine-treated cells (for 3 hr) stained for DNA (blue), Ndc80 (red) and tubulin (green) showing that RNAi against *Sf3A2* or *Prp31* does not affect Ndc80 accumulation at kinetochores.
DOI: https://doi.org/10.7554/eLife.40325.031
**Figure supplement 3.** Biochemical relationships between Ndc80, Nuf2, Sf3A2 and Prp31.
DOI: https://doi.org/10.7554/eLife.40325.032

*Figure 7D*). Moreover, the kinetochores of colchicine-treated (for 3 hr) *Sf3A2* and *Prp3* RNAi cell displayed the same levels of Ndc80 as colchicinized control cells (*Figure 7—figure supplement 2*). This finding suggests that Ndc80 association with kinetochores of SF-depleted cells is destabilized in the presence of the spindle MTs. Comparable results were obtained with Spc25/Mitch. Mitch accumulation at kinetochores was substantially reduced in *Sf3A2* and *Prp3* RNAi cells not treated with colchicine, but not in RNAi cells exposed to the drug (*Figure 7B*). Although we could not assess whether Sf3A2- and Prp31-depleted cells contain normal amounts of Spc25/Mitch, because the anti-Mitch antibody works well in indirect immunofluorescence but not in Western blotting (*Williams et al., 2007*), the Mitch protein levels were not significantly reduced in our whole proteome analyses. We could not test whether Nuf2 is mislocalized in the SF-depleted cells, because our antibody against this protein does not work in indirect immunofluorescence. However, western blotting showed that SF-depleted cells exhibit normal levels of Nuf2, in agreement with the results of our proteomic analyses (*Figure 6*, *Supplementary file 1*), *Figure 7D*).

We next asked whether Sf3A2 and Prp31 interact physically with the components of the Ndc80 complex. In humans, the Ndc80 complex consists of four subunits (Ndc80/HEC1, Nuf2, Spc24 and Spc25); Ndc80/HEC1 and Nuf2 form a tightly packed dimer and their distal calponin-homology domains promote MT association with the kinetochore (reviewed in *DeLuca and Musacchio, 2012*; *Varma and Salmon, 2012*). Ndc80/HEC1 and Nuf2 are mutually dependent for their in vivo stability, as RNAi against either protein results in reduction of both proteins (*DeLuca et al., 2003*). However, the entire Nuf2 protein and very large Ndc80/HEC1 fragments are stable when expressed in bacteria in the absence of Ndc80/HEC1 and Nuf2, respectively (*Ciferri et al., 2005*; *Wu et al., 2009*). In *Drosophila* living cells, RNAi against *Ndc80* leads to a strong Nuf2 reduction, while RNAi against *Nuf2* slightly reduces the Ndc80 level (*Figure 7—figure supplement 3A*; see also (*Przewloka et al., 2007*). We also found that full-length *Drosophila* Ndc80 and Nuf2 are stable when are singly expressed in bacteria as GST-tagged proteins (*Figure 7E*, *Figure 7—figure supplement 3*).

To detect physical interactions between the SFs and the Ndc80 complex, we first tested by Co-IP whether the SFs co-precipitate with Ndc80 and Nuf2. Reciprocal Co-IP assays showed that each splicing factor interacts in vivo with both Ndc80 and Nuf2 (*Figure 7—figure supplement 3C*). These results were confirmed by GST pulldown experiments from S2 cells extracts expressing 3HA-Sf3A2 or 3HA-Prp31; we found that GST-Ndc80, GST-Nuf2 and GST-Mitch efficiently precipitate 3HA-Sf3A2 and 3HA-Prp31 (*Figure 7—figure supplement 3B*). We then asked whether the Co-IP results reflect direct protein-protein interactions and performed GST pulldown experiments using bacterially expressed purified proteins (6His-Sf3A2, 6His-Prp31, GST-Ndc80, GST-Nuf2 and GST-Mitch). We carried out pulldowns with glutathione-agarose beads bound to GST-Ndc80, GST-Nuf2, or GST-Mitch. Both 6His-Sf3A2 and 6His-Prp31 efficiently bound the GST-tagged components of the Ndc80 complex but not GST alone (*Figure 7E*). To check for a non-specific protein-binding activity of the His-tagged SFs we tested whether they bind two GST-tagged proteins functionally unrelated with kinetochores; we found that both SFs fail to bind the *Drosophila* telomeric proteins HOAP (HP1/ORC-associated protein) and Ver (Verrocchio) (*Raffa et al., 2011*), arguing against the possibility of nonspecific interactions between the SFs and the Ndc80 complex (*Figure 7—figure supplement 3D*). Finally, given that Sf3A2 and Prp31 bind the same kinetochore components, we asked whether they interact with each other. Reciprocal Co-IP experiments from S2 cell extracts clearly showed that they co-purify (*Figure 7F*).

## The phenotypes of Ndc80- and SF-depleted cells are not identical

The finding that Sf3A2 and Prp31 interact with Ndc80 and facilitate its binding to the kinetochore prompted us to compare the phenotypes of Ndc80- and SF-depleted cells (*Figure 8*). Fixed *Ndc80* RNAi cells showed many PMLES (26%) and a reduced frequency of ana-telophases just like the SF-depleted cells. However, in addition to PMLES, Ndc80 RNAi cells also exhibited ana-telophase-like figures with decondensed chromosomes. These latter figures were not stained by either Cyclin B or BubR1 and almost invariably showed unequal chromosome segregation (*Figure 8B, C and D*). How-ever, we could not assess whether they contained chromosomes with unseparated sister chromatids. The in vivo mitotic phenotype of Ndc80-depleted cells was in part different from that elicited by loss of Sf3A2 or Prp31. In 14 cells filmed, four entered anaphase within 45 min; in the remaining 10 cells, following an initial clustering at the center of the cell, the chromosomes exhibited rapid oscillations and eventually spread out throughout the cell. However, after approximately 90 min of oscillations, chromosomes started to decondense as normally occurs at the end of telophase (*Figure 8E*, *Video 9*). A similar chromosome decondensation process was not observed in cells depleted of either Sf3A2 or Prp31.

To further investigate the functional relationships between the SFs and Ndc80, we asked whether the splicing-independent overexpression of Ndc80 rescues the phenotype elicited by *Sf3A2* or *Prp31* RNAi. We used an S2 cell line bearing a construct containing the Tap-tagged *Ndc80* cDNA under the control of the copper-inducible metallothionein (Mtn) promoter. The tagged Ndc80 pro-tein localizes normally to the kinetochores and can be overexpressed using high concentrations of copper sulfate (*Figure 8F*). These cells were subjected to RNAi against the SFs and simultaneously treated with 20 or 100 mM copper sulfate to induce relatively low (L) and high (H) expression levels of Ndc80, respectively (*Figure 8F*). Cytological analysis showed that Ndc80-TAP expression at low level does not alter the mitotic phenotypes observed after Sf3A2 or Prp31 depletion (*Figure 8D*). High expression levels of Ndc80-TAP slightly increased the frequencies of ana-telophases but did not affect the PMLES frequencies (*Figure 8D*). These results indicate that the mitotic phenotypes eli-cited by RNAi against the SFs are largely independent of the intracellular levels of Ndc80.

## The human homologues of Sf3A2 and Prp31 play a conserved mitotic function in HeLa cells

Lastly, we asked whether the mitotic roles of Sf3A2 and Prp31 are conserved in human cells. We first determined the localization of these proteins by immunostaining dividing HeLa cells with antibodies directed to the human orthologues of Sf3A2 and Prp31. This localization was similar to that seen in *Drosophila* S2 cells (see *Figure 5A*, *Figure 9*, *Figure 9—figure supplement 1*); the human SFs were present in the nucleus during interphase, while in metaphases with tightly aligned chromosomes they were associated with the central part of the spindle. During anaphase, the SFs localized to the center of the cell, while in telophase they were included in the reforming nuclei (*Figure 9A*, *Fig-ure 9—figure supplement 1*). The staining of metaphase spindles with anti-SF3A2 or anti-PRP31 antibodies was not observed after siRNA-mediated depletion of these factors (*Figure 9A and B*). Brief nocodazole treatments led to a diffuse localization of the SFs in metaphase cells, suggesting that they associate with the spindle and not with the spindle matrix, consistent with the results obtained in S2 cells (*Figure 9A*).

We next examined the phenotypes of HeLa cells after RNAi against *SF3A2* and *PRP31*. Both genes were silenced efficiently leading to very strong reductions of the corresponding proteins (*Figure 9B*). Loss of either SF3A2 or PRP31 resulted in a virtually identical mitotic phenotype, com-parable to that observed in *Drosophila* cells depleted of the homologous SFs. We observed cells with morphologically irregular spindles, a strong reduction in the normal frequency of metaphases with congressed chromosomes, frequent monopolar spindles, and a low frequency of ana-telophases (*Figure 9C*). These phenotypes were rescued by the ectopic expression of FLAG-tagged *SF3A2* or *PRP31* genes bearing translationally silent mutations that render them insensitive to the siRNAs (*Figure 9C*).

We next analyzed the localization of HEC1, the human protein orthologous to Ndc80, on kineto-chores of SF3A2 and PRP31 depleted cells. We first examined cells not treated with nocodazole and fixed with a standard paraformaldehyde-based procedure (see Materials and methods) stained with an anti-HEC1 antibody. Consistent with the results in *Drosophila*, HEC1 accumulation at the

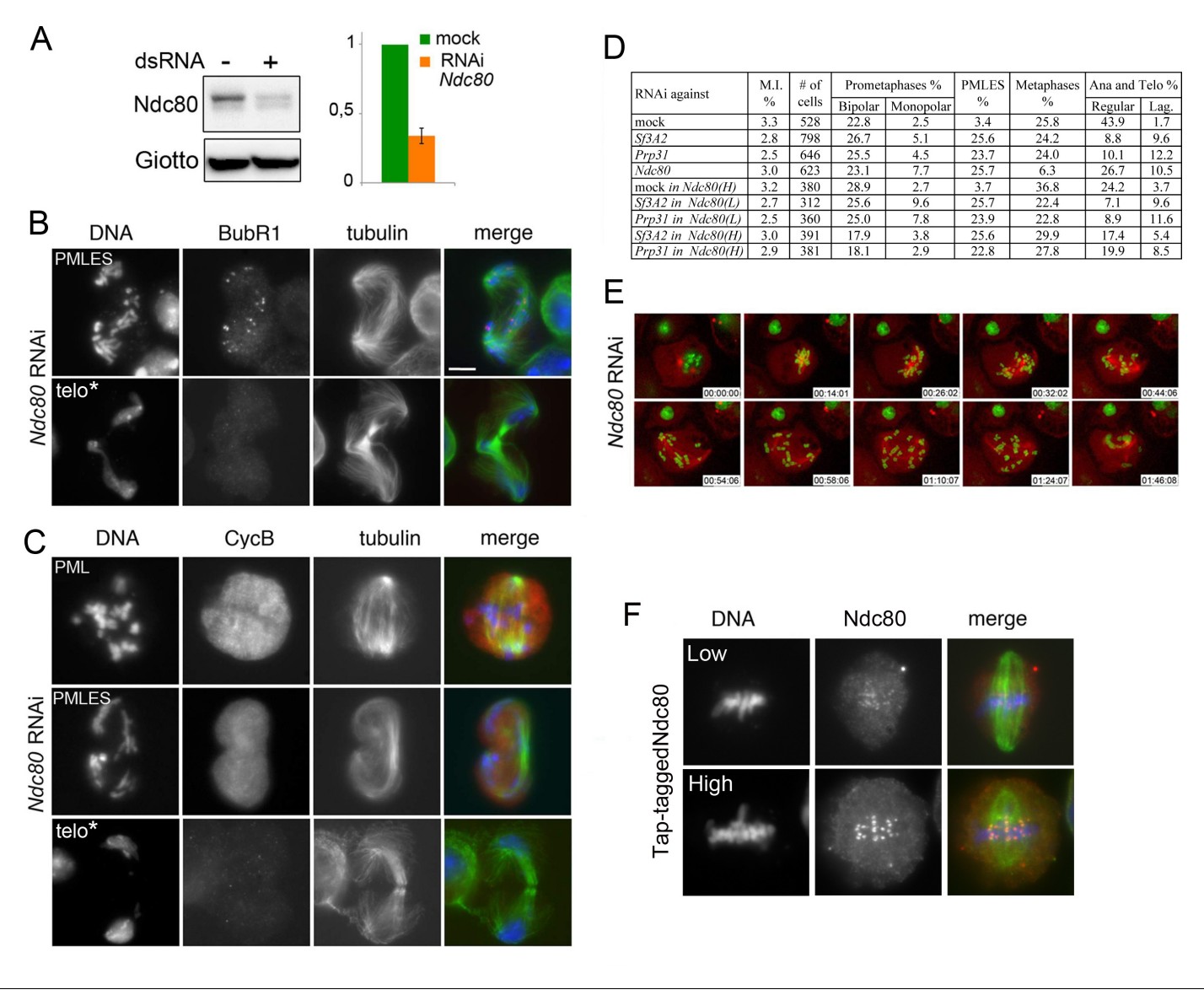

**Figure 8.** Phenotype of Ndc80-depleted cells. (A) Western blots of S2 cell extracts and quantitation of relative band intensities showing that RNAi against *Ndc80* strongly reduces the level of this protein. Giotto is a loading control. (B and C) Mitotic figures observed in Ndc80-depleted cells stained for DNA (blue), tubulin (green), BubR1 and Cyclin B (red). PML, prometaphase-like; PMLES, prometaphase-like cells with elongated spindles; telo*, telophase-like figures with decondensed chromosomes that might represent stages subsequent to PMLES. Note that PMLES exhibit BubR1-stained kinetochores and a high level of Cyclin B, whereas telo* figures no longer exhibit BubR1 and Cyclin B signals. See text for detailed explanation. Scale bars, 5 μm. (D) Mitotic indexes (M.I.) and frequencies of mitotic figures observed after RNAi against *Sf3A2*, *Prp31* or *Ndc80*. Where indicated, RNAi against *Sf3A2* or *Prp31* was performed in cells that express Tap-tagged *Ndc80* cDNA under the control of the copper-inducible Mtn promoter. Ndc80 (L) and Ndc80 (H) indicate low and high Ndc80-Tap expression. Lag, lagging chromosomes between the ana-telophase nuclei. Data on mock-treated control cells and SF-depleted cells are the same as those of *Figure 1D*. (E) Stills from time-lapse videos of mitosis in mock-treated and *Ndc80* RNAi cells expressing histone-GFP and cherry-tubulin. The numbers at the bottom of each frame indicate the time elapsed from the beginning of imaging (h: min:s) See text for description of chromosome behavior. See also *Videos 1* and *9*. (F) Examples of cells expressing Tap-tagged Ndc80 induced by 20 mM (low) and 100 mM (high) copper sulfate. Note that that Tap-tagged Ndc80 accumulates normally at kinetochores.
DOI: https://doi.org/10.7554/eLife.40325.034

The following source data is available for figure 8:

**Source data 1.** Source data for *Figure 8*.
DOI: https://doi.org/10.7554/eLife.40325.035

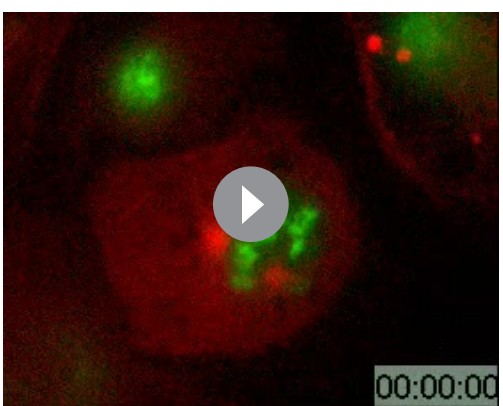

**Video 9.** Metaphase delay in Ndc80-depleted S2 cells expressing histone-GFP and tubulin mCherry.
DOI: https://doi.org/10.7554/eLife.40325.036

kinetochores was reduced, without a concomitant significant reduction in the amount of the protein (*Figure 9B and D*). We specifically observed that in most RNAi cells there was a halo of HEC1 in the cytoplasmic area surrounding the chromosomes. We measured the fluorescence of this halo in prometaphase cells, and found that it was significantly more intense in RNAi cells than in control cells (*Figure 9D*). *SF3A2* and *PRP31* RNAi cells treated with nocodazole (for 3 hr) displayed HEC1 signals comparable to those of control cells and most of them did not show a fluorescent HEC1 halo around the chromosomes (*Figure 9E*).

This similarity between the HeLa and the *Drosophila* S2 mitotic phenotype prompted us to ask whether HEC1 interacts with SF3A2 and PRP31. We performed reciprocal Co-IP experiments using HeLa cell extracts. Consistent with the *Drosophila* results, we found that both SFs physically interact with each other and with HEC1 (*Figure 10A and B*). We also sought to determine when during the cell cycle HEC1 interacts with the SFs. We could not perform this experiment in S2 cells because they cannot be synchronized. We synchronized HeLa cells using a double thymidine block and performed Co-IP from whole cell extracts with an anti-HEC1 antibody at various times after release from the block. The anti-HEC1 antibody failed to precipitate either SF3A2 or PRP31 from cells in the S phase (time 0) or in the G2 phase (5 hr after release of the block), but efficiently pulled down both SFs from dividing cells expressing the mitotic marker phospho-histone H3 (PHH3) (*Goto et al., 1999*) 8 hr after release from the block (*Figure 10C*).

To obtain further insight into the interaction between HEC1 and the SFs, we synchronized HeLa cells with a 16 hr nocodazole treatment; mitotic cells were then collected by mechanical shake-off both at the end of nocodazole treatment (0 hr time) and 6 hr after the drug removal. These cells populations were both highly enriched in cells undergoing mitosis marked by the PHH3, which was not found in adherent cells (*Figure 10D*). Co-IP with an anti-HEC1 antibody did not detect any HEC1-SF interaction in cells shaken-off from nocodazole-containing cultures or from adherent cells. However, mitotic cells shaken off from cultures kept in fresh medium for 6 hr after nocodazole removal displayed a robust interaction between HEC1 and either SF3A2 or PRP31. These results indicate that the physical interaction between HEC1 and either SF3A2 or PRP31 is restricted to mitosis, when pre-mRNA splicing is suppressed (*Hofmann et al., 2010*; *Shin and Manley, 2002*) and kinetochore proteins and SFs can come into contact following nuclear envelope breakdown. The finding that this interaction does not occur in mitotic cells with depolymerized MTs, but takes place when these cells have reassembled a spindle, further suggests that the HEC1-SF interaction requires the presence of the spindle MTs.

To further characterize HEC1 association with kinetochores after *SF3A2* or *PRP31* siRNA-mediated depletion, we fixed cells using a procedure that is thought to improve visualization of proteins associated with kinetochore MTs (fixation after pre-extraction with Triton X-100; see, for example, *Roscioli et al., 2012*; *Tanenbaum et al., 2006*). Prometaphase and metaphase cells fixed in this way exhibit chromosome-associated MT bundles that are often bent and tangled and no longer exhibit the typical spindle-like structure. SF3A2- and PRP31-depleted cells fixed with this procedure were co-immunostained with an anti-HEC1 antibody and either an anti-tubulin or a CREST antibody that marks the kinetochores. In contrast to control cells, in which HEC1 staining was confined to kinetochores, in SF3A2- or PRP31-depleted mitotic cells, we found HEC1 associated with filaments that in many cases appear to protrude from the CREST-stained kinetochores (*Figure 11A*); these filaments were also stained by anti-tubulin antibodies, suggesting that they are kinetochore fibers (*Figure 11B*). We noted a variability in this phenotype; in some cases, the bulk of HEC1 was still bound to kinetochores and only a fraction of the protein was associated with the filaments; in other cases a substantial part of HEC1 was no longer on kinetochores and was found on the tubulin filaments. (*Figure 11B*). We also stained Triton X-100 pre-extracted cells with antibodies against HURP,

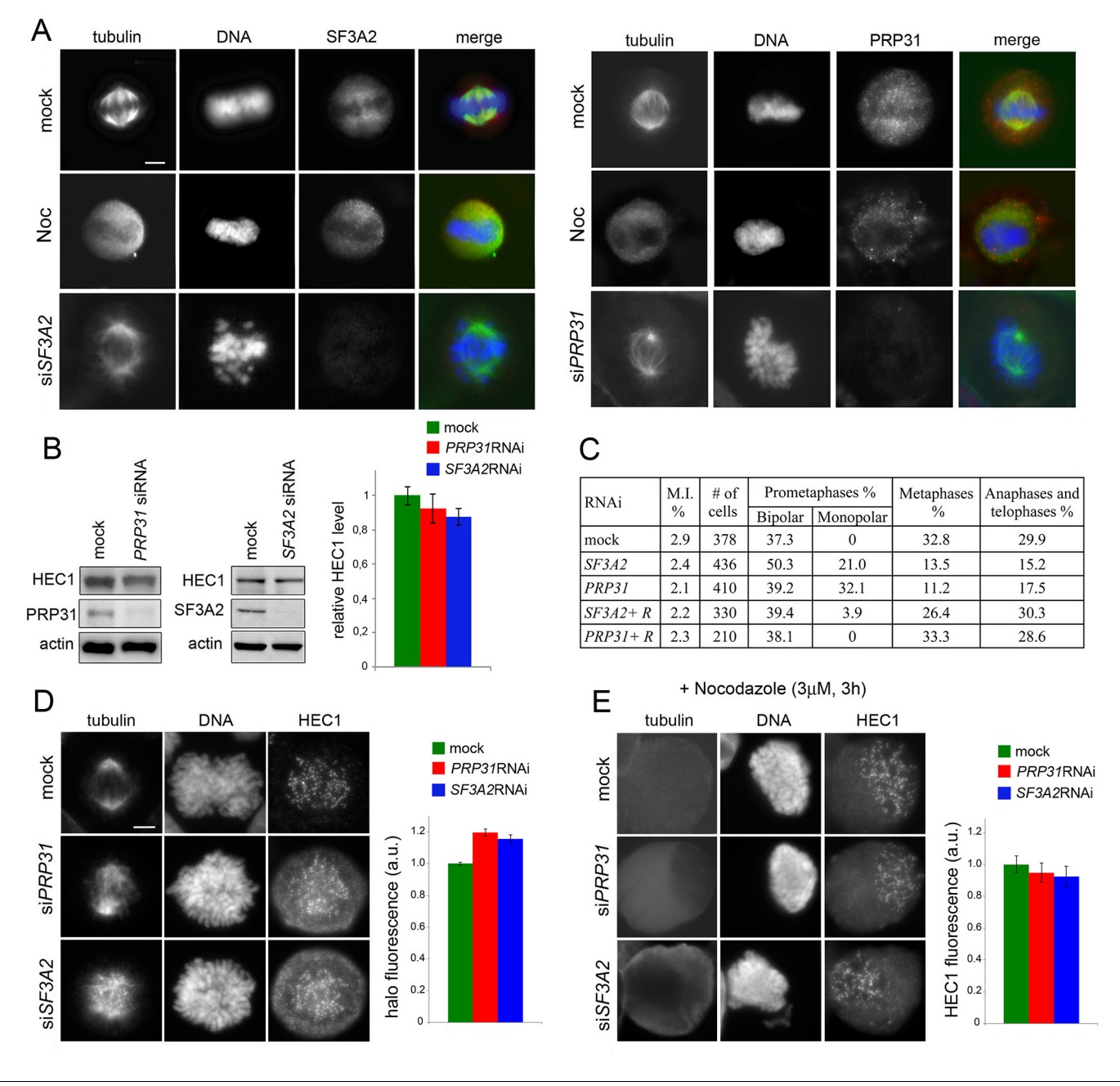

**Figure 9.** SF3A2 and PRP31 interact with HEC1/Ndc80 and are required for chromosome segregation in HeLa cells. (**A**) SF3A2 and PRP31 localization in HeLa cells; both SFs are slightly enriched in the central part of the spindle region; in nocodazole (Noc)-treated cells the SFs are diffuse in the mitotic cytoplasm; in *SF3A2* or *PRP31* RNAi cells, the spindles are not immunostained by the pertinent antibodies. In merged images, DNA is blue, tubulin green and the SFs red. Scale bar, 5 μm. (**B**) Western blotting showing that RNAi-mediated depletion of SF3A2 or PRP31 strongly reduces the levels of these proteins without affecting HEC1 expression. Actin is a loading control. Quantification of the HEC1 level from three independent experiments does not reveal significant differences between control and RNAi cells using unpaired t-test. Error bars, SEM. (**C**) Mitotic index (M.I.) and frequencies of mitotic figures observed after RNAi against *SF3A2* and *PRP31*. R, are specific rescue constructs expressing RNAi resistant *SF3A2* and *PRP31* genes. (**D**) Representative prometaphase figures (not treated with nocodazole) stained for DNA, HEC1 and tubulin showing that RNAi against SF3A2 or PRP31 results in the formation of a HEC1 halo surrounding the chromosomes. The graph shows quantification of halo fluorescence. To estimate the fluorescence intensities of the halos, we calculated the signal-to-background ratios in 20 randomly chosen cells, as described in Materials and methods. In SF3A2 and PRP31 RNAi cells, these ratios are significantly higher than that detected in control cells (p<0.001 for both samples in the unpaired t test). (**E**) Representative nocodazole-treated mitotic figures stained for tubulin and HEC1. The graph shows a quantification of the HEC1 fluorescence

*Figure 9 continued on next page*

*Figure 9 continued*

associated with the chromosomes; see Materials and methods for details on fluorescence measurement. The fluorescence intensities detected in *SF3A2* and *PRP31* RNAi cells are not significantly different from that of control cells in the unpaired t test.

DOI: https://doi.org/10.7554/eLife.40325.037

The following source data and figure supplement are available for figure 9:

**Source data 1.** Source data for *Figure 9B,D and E*.

DOI: https://doi.org/10.7554/eLife.40325.039

**Figure supplement 1.** SF3A2 localization in HeLa cells Immunolocalization of SF3A2 (red) during mitosis of HeLa cells stained for tubulin (green) and DNA (blue).

DOI: https://doi.org/10.7554/eLife.40325.038

which localizes predominantly to the kinetochore fibers (*Silljé et al., 2006*) and found that the HEC1 filaments observed in SF3A2 and PRP31-depleted cells are enriched in HURP (*Figure 11—figure supplement 1*).

## Discussion

### The phenotypic consequences of Sf3A2 or Prp31 deficiency are SAC-dependent

We have shown that depletion of either Sf3A2 or Prp31 disrupts mitotic chromosome segregation in *Drosophila* embryos and somatic cells. In *Drosophila* S2 cells and brains, depletion of either SF resulted in metaphase delay, failures in chromosome segregation and frequent monopolar spindles; a mitotic phenotype that is consistent with defective k-fiber formation and/or function (*Somma et al., 2008*; *Toso et al., 2009*). We also showed that SF depletion in S2 cells prevents SAC satisfaction, contributing to the metaphase arrest phenotype. Notably, although S2 cells depleted of either *Sf3A2* or *Prp31* are both defective in Ndc80 localization at kinetochores, their phenotypes detected in living cells were rather different from that elicited by Ndc80 depletion. Following RNAi against *Sf3A2* or *Prp31*, most cells remained arrested in metaphase for long times without showing substantial chromosome decondensation. In contrast, most *Ndc80* RNAi cells were delayed in a prolonged prometaphase-like state but, after many oscillations, their chromosomes decondensed as occurs during normal telophase. We believe that this phenotypic diversity depends on differences in both kinetochore function (see below) and SAC activity. The chromosome decondensation phenotype observed in Ndc80-depleted cells could reflect a partial impairment of the SAC, which is known to be attenuated in S2 cells lacking Ndc80 (*Feijão et al., 2013*). Thus, some of the Ndc80-depleted cells would proceed through the mitotic process and undergo chromosome decondensation. In *Sf3A2* and *Prp31* RNAi cells, even if Ndc80 is abnormally distributed, the SAC machinery is probably still functional but the SAC is not satisfied, leading to metaphase arrest and preventing mitotic progression and chromosome decondensation.

Injections of either α-SfA3 or α-Prp31 into *Drosophila* embryos caused very similar mitotic defects, although these mitotic phenotypes differ from those seen in S2 or larval brain cells. Inhibition of either SF in syncytial embryos led to spindles which initially form normally, but then reduce in size, leading to short metaphase-like spindles with poorly aligned chromosomes. These spindles, however, exit mitosis after a delay, with anaphase-like MT dynamics but a catastrophic failure of chromosome segregation, followed by chromosome decondensation. These results indicate that in *Drosophila* embryos Sf3A2 and Prp31 are required for proper kinetochore-MT interactions and chromosome segregation. In addition, because chromosomes of colchicine-treated or anti-Dgt6 (augmin) injected embryos arrest in metaphase for several minutes without undergoing decondensation (*Hayward et al., 2014*; *Zalokar and Erk, 1976*), we further suggest that inactivation of Sf3A2 or Prp31 affects SAC function in embryos. Interestingly, a phenotype similar to that elicited by anti-Sf3A2 or anti-Prp31 antibody injection was observed in embryos homozygous for mutations in the *mis12* or *Kmn1* (formerly *Nsl1*) genes that encode kinetochore components whose loss partially impairs the SAC (*Venkei et al., 2011*). It is also possible that premature chromosome decondensation in early anaphase of αSf3A2 and αPrp31 injected embryos is a consequence of their delayed progression through metaphase. Finally, the phenotypic differences between embryonic and S2 cells

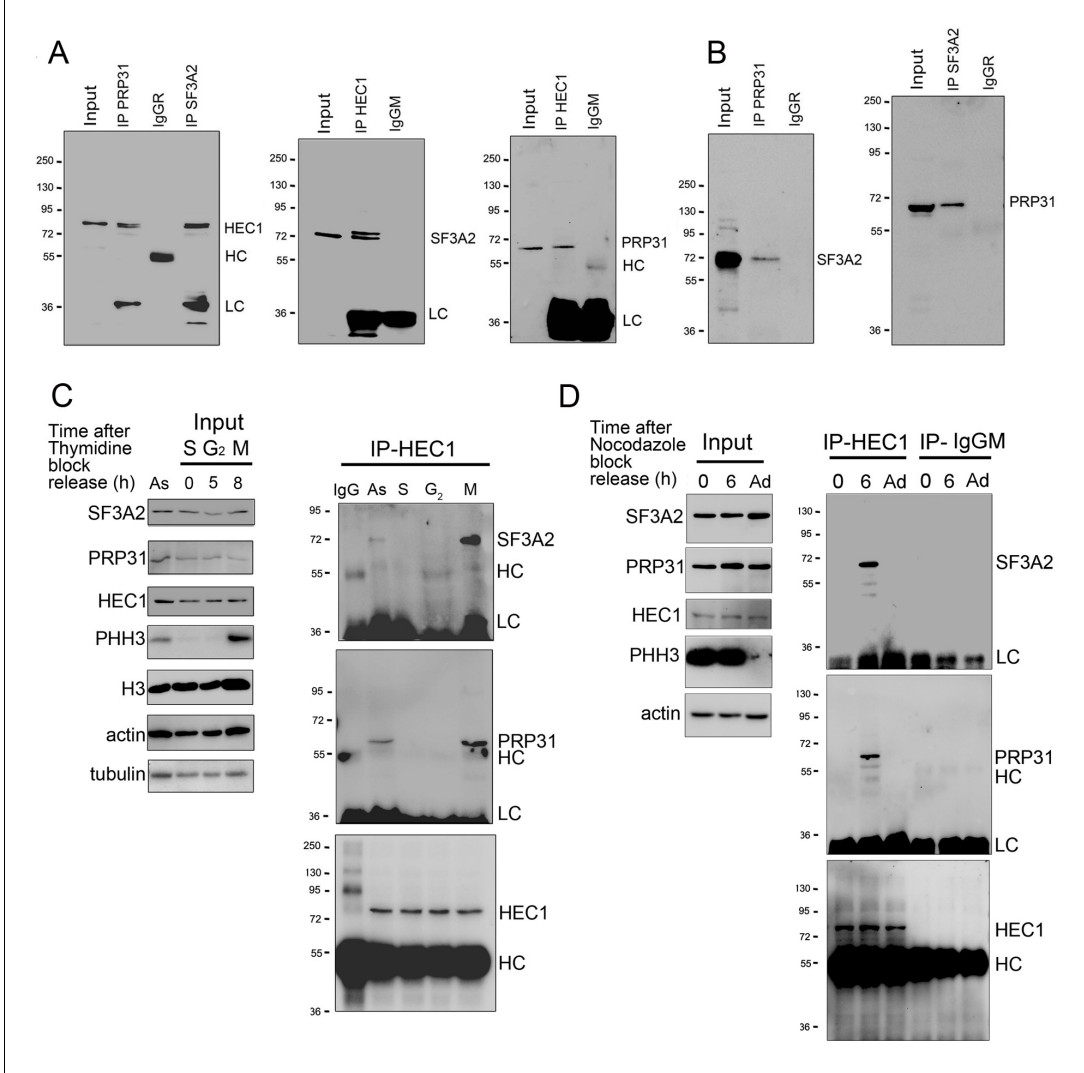

**Figure 10.** SF3A2 and PRP31 interact with HEC1 only during the M phase in the presence of MTs. (**A**) Co-IP analysis showing that SF3A2 and PRP31 interact with HEC1 (input was 8% in all experiments). HC and LC, IgG heavy and light chains, respectively. (**B**) Reciprocal Co-IPs showing that SF3A2 and PRP31 interact with each other. (**C**) HEC1-SF interaction in HeLa cells synchronized using a double thymidine block. Cells were examined at different times after release from the block (see Materials and methods for details). As, Asynchronous cells; S (0 hr), G2 (5 hr) and M (8 hr) indicate the phases of the cell cycle; PHH3 is the mitotic marker phosphohistone H3; H3, histone H3; LC and HC are the IgG light and heavy chains. Tubulin and actin are loading controls. Note that SF3A2 and PRP31 coprecipitate with HEC1 only when the sample contains mitotic cells. (**D**) SF-interaction in HeLa cells synchronized using nocodazole treatment. Cells were collected by mitotic shake-off at 0 hr or 6 hr after nocodazole removal; Ad, Adherent cells. Note that SF3A2 and PRP31 do not interact with HEC1 in cells with depolymerized MTs (0 hr).

DOI: https://doi.org/10.7554/eLife.40325.040

observed after Sf3A2 or Prp31 inhibition might reflect differences in the SAC mechanisms operating in the two cell types, as has been previously described (*Blower et al., 2006*; *Venkei et al., 2011*).

## Sf3A2 and Prp31 play direct mitotic roles conserved from flies to humans

Our findings on *Drosophila* and human cells strongly suggest that Sf3A2 and Prp31 play direct mitotic roles independent of their roles in pre-mRNA splicing. Our analyses on S2 cells have produced several results that support this conclusion. First, we have shown that Sf3A2 or Prp31 deficiency produces strong mitotic effects that are not elicited by depletion of other factors with which they form splicing subcomplexes, a finding that suggests the Sf3A2 and Prp31 have specific and

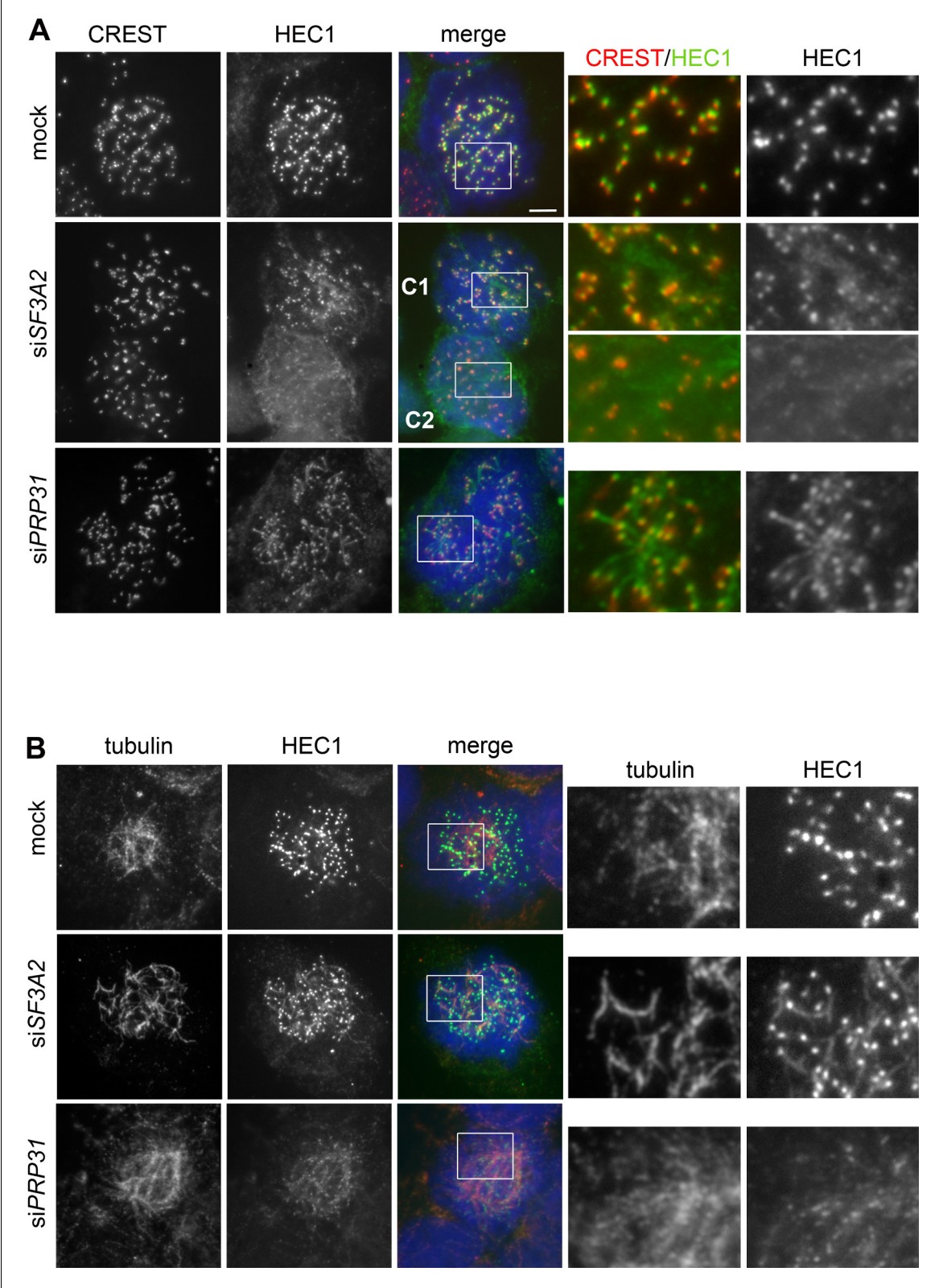

**Figure 11.** SF3A2 and PRP31 are required to prevent diffusion of HEC1 from kinetochores of HeLa cells. (**A, B**) Prometaphases and prometaphase-like figures were fixed after pre-extraction with Triton X-100 (see Materials and methods) and stained with DAPI (DNA, blue) and either CREST (red) and anti-HEC1 (green) antibodies (**A**) or anti-tubulin (red) and anti-HEC1 (green) antibodies (**B**). Boxes delimit regions shown as enlargements in the rightmost panels. In *SF3A2* and *PRP31* RNAi cells the anti-HEC1 antibodies stain filaments protruding from the CREST signals (**A**); these filaments are

*Figure 11 continued on next page*

*Figure 11 continued*

also stained by anti-tubulin antibodies (**B**). The middle panels in A show two cells, C1 and C2. In C1, HEC1 is mostly associated with the kinetochores, showing only short protrusions from the CREST signals; in C2, the bulk of HEC1 is no longer on kinetochores but is instead associated with filamentous structures. Scale bar, 5 μm.

DOI: https://doi.org/10.7554/eLife.40325.041

The following figure supplement is available for figure 11:

**Figure supplement 1.** Localization of HEC1 in HeLa cells depleted of SF3A2 and PRP31 and stained for HURP.

DOI: https://doi.org/10.7554/eLife.40325.042

possibly direct mitotic roles. Second, our proteome-wide analysis of S2 cells showed that RNAi against *Sf3A2* or *Prp31* causes a significant reduction in only a small proportion of the proteome. Of these proteins, all except one (HUS1-like) are also reduced in Sf3A1 RNAi-treated cells that do not exhibit any mitotic phenotype. As HUS1-like is involved in homologous recombination during meiosis (*Peretz et al., 2009*), it is highly unlikely that its depletion results in the mitotic defects observed in *Sf3A2*- or *Prp31*-depleted cells. Third, we have shown that *Drosophila* Sf3A2 and Prp31 physically interact with both the MTs and the components of the Ndc80 complex, an additional feature suggesting a direct participation of these factors in the mitotic process.

Further strong evidence for a direct mitotic role of Sf3A2 and Prp31 is also provided by the results of our embryo injection experiments. The mitotic phenotype observed in α-Sf3A2- and α-Prp31-injected embryos is quite specific and was not previously observed after injection of antibodies against several different proteins enriched in chromatin and/or the mitotic apparatus, such as Skeletor (*Walker et al., 2000*), Cid/Cenp-A (*Blower and Karpen, 2001*), mitotic motors (cytoplasmic dynein, Klp61F or Ncd) (*Sharp et al., 2000*), Eb1 (*Rogers et al., 2002*), and the Akt kinase (*Buttrick et al., 2008*). Thus, the mitotic defects observed in embryos appear to be the consequence of loss of a specific mitotic function rather than the result of a general disturbance in embryo development.

Early *Drosophila* embryos undergo 13 extremely rapid nuclear divisions within a syncytium. These divisions predominantly rely on already spliced mRNA molecules that are packaged into the egg by the mother, although a minor wave of transcription has been reported during cycles 8–13 (*Tadros and Lipshitz, 2009*). Although it could be argued that Sf3A2 and Prp31 function in the syncytium in splicing this small number of transcripts, three pieces of evidence refute this. First, the genes that are expressed early in embryogenesis are small and generally lack introns (*De Renzis et al., 2007*). Second, the few early transcripts with introns require the Fandango/Xab2 subunit of the NCT/Prp19 complex for splicing. However, *fandango* mutants exhibit normal syncytial mitoses and show abnormal nuclear shaping only upon cellularization, during the 14th cell cycle (*Guilgur et al., 2014*). Third, the mitotic phenotypes seen upon α-Sf3A2 or α-Prp31 injection occur less than 1 min following injection. In addition, the bulk of the injected antibody penetrates into the nucleus only when it becomes fenestrated at early prometaphase. Thus, injected antibodies can bind and inhibit the SFs only when dividing embryonic nuclei are in prometaphase. To postulate that the mitotic defect observed in injected embryos is not a direct effect of the SF inhibition but is instead a consequence of failed pre-mRNA processing one has to make three highly unlikely assumptions: (i) that mRNA splicing in *Drosophila* embryos occurs during mitosis when transcription and splicing are usually suppressed (*Hofmann et al., 2010*), (ii) that splicing and translation of a hypothetical mitotic pre-mRNA, and the folding and activation of its downstream protein all takes less than 60 s, a time interval that does not appear to be sufficient for these processes (*Beyer and Osheim, 1988*; *Huranová et al., 2010*; *Khodor et al., 2011*), and (iii) that the mitotic protein whose mRNA splicing is acutely inhibited by antibody injection has an extremely rapid turnover and needs to be continuously replenished throughout cycles 10–13.

Our conclusion that Sf3A2 and Prp31 have direct mitotic functions is strongly supported by our results in human cells. Indeed, we have shown that HeLa cells depleted of SF3A2 or PRP31 exhibit mitotic phenotypes similar to those observed in *Drosophila* S2 cells and larval brains. In addition, HEC1/Ndc80 physically interacts with SF3A2 and PRP31 also in HeLa cells, and the HEC1-SF interaction is restricted to the M phase and is MT-dependent. These results strongly suggest that the functions and behavior of Sf3A2 and Prp31 are evolutionarily conserved. These findings strengthen our

view that these SFs play direct mitotic roles, and also suggest that these roles are ancient and might therefore be common to many organisms.

## What are the direct mitotic roles Sf3A2 and Prp31?

Although our data indicate that Sf3A2/SF3A2 and Prp31/PRP31 have direct mitotic functions, these functions are not well defined. We have shown that in *Drosophila* and human cells Ndc80/HEC1 physically interacts with Sf3A2/SF3A2 and Prp31/PRP31, and that in HeLa cells the physical interaction between HEC1 and the SFs is MT-dependent and does not occur in mitotic cells with depolymerized MTs. In Sf3A2- and Prp31-deficient *Drosophila* and human cells fixed with formaldehyde (without Triton X-100 pre-extraction) Ndc80/HEC1 failed to accumulate properly at kinetochores and displayed different degrees of dispersion in the mitotic cytoplasm. However, when these SF-depleted cells were treated with colchicine or nocodazole, they displayed normal amounts of kinetochore-associated Ndc80/HEC1.

In SF-depleted HeLa cells fixed after pre-extraction with Triton X-100, HEC1 was partially diffused along the remnants of the k-fibers. A similar HEC1 diffusion along MTs was not observed in control cells. We note that we were not able to obtain a similar result in *Drosophila* because the pre-extraction procedure used for HeLa cells disrupts the morphology of *Drosophila* mitotic cells. The Ndc80 complex interacts with the spindle MTs mainly through its Ndc80/HEC1 subunit and constitutes the major kinetochore-MT binding interface (reviewed in *DeLuca and Musacchio, 2012*; *Varma and Salmon, 2012*). The complex forms discrete oligomeric linear arrays along the kinetochore MTs. These arrays have the ability to diffuse along the MTs and are thought to have a stabilizing effect on kinetochore MTs (*Alushin et al., 2010*). Collectively, these results suggest a model for the mitotic role of SF3A2 and PRP31. We propose that these SFs interact with Ndc80/HEC1 on the k-fibers, regulating the association of the Ndc80 complex with the kinetochore. In the absence of the SFs, the Ndc80/HEC1 complex would associate with the kinetochores, or the kinetochore-MT interfaces, in an abnormal fashion, so that the binding of the complex to the kinetochore would be weakened and the SAC satisfaction prevented. As a result, when the kinetochores are attached to MTs, Ndc80/HEC1 would diffuse away from the kinetochores possibly along the kinetochore fibers. However, although the most straightforward interpretation of our observations on Triton X pre-extracted cells is that HEC1 spreads out along the kinetochore MTs, we cannot exclude the possibility that the association of HEC1 with the spindle MTs depends on the fixation technique. It is indeed possible that HEC1 dissociates from the kinetochores and then binds the MTs that resisted the pre-extraction/fixation procedure. Intriguingly, several features of the SF-Ndc80/HEC1-MT interactions are reminiscent of those described for the mammalian Ska complex, which is required for proper kinetochore- MT-attachment and SAC satisfaction (*Auckland et al., 2017*; *Hanisch et al., 2006*). The Ska complex directly interact with the Ndc80 complex and with microtubules and this interaction is required for Ska localization at the kinetochores (*Helgeson et al., 2018*; *Janczyk et al., 2017*). The main difference between the Ska and the SF behavior is that Ska accumulates at kinetochores while the SFs do not appear to have this ability.

Recent work using *Xenopus* egg extracts has shown that inhibition of RNA splicing leads to the accumulation of unprocessed centromeric noncoding RNAs (ncRNAs) accompanied by decreased recruitment of centromere/kinetochore proteins such as CENP-A, CENP-C and NDC80 (*Grenfell et al., 2016*). However, the persistence of these centromere-associated RNAs does not appear to be required for normal kinetochore assembly (*Grenfell et al., 2017*). It has been instead suggested that a transient association of RNAs with the centromere stabilizes CENP-C and contributes to the recruitment of SFs, which could play important regulatory roles at kinetochores (*Grenfell et al., 2017*). It has been also shown that non-coding transcripts from the 359 bp *Drosophila* satellite DNA localize to the centromeric regions of the chromosomes and are required for the correct localization of the fly homologues of CENP-A (Cid) and CENP-C (*Rošić et al., 2014*). Moreover, recent results strongly suggest that transcription occurs at S2 cell centromeres and it is necessary for Cid incorporation into centromeric chromatin (*Bobkov et al., 2018*). Our findings are compatible with the studies on *Xenopus* extracts and *Drosophila* and do not exclude that Sf3A2 and Prp31 could transiently interact with centromere/kinetochore-non coding associated RNAs. We have shown that these SFs bind both MTs and the Ndc80 complex, two events that could facilitate their interaction with centromeric RNAs.

In conclusion, we have shown that Sf3A2 and Prp31 and their human orthologues are required for mitotic division. Our data argue strongly against the possibility that the mitotic defects caused by Sf3A2 or Prp31 inhibition are the consequence of a failure to splice protein-coding mRNAs. We therefore propose that Sf3A2 or Prp31 play a direct and evolutionarily conserved role in open mitosis.

## Materials and methods

### Cell culture and RNA interference

S2 cells (DGRC, RRID:CVCL_Z232; tested negative for mycoplasma) were cultured at 25°C in Schneider's insect medium (Sigma) supplemented with 10% heat-inactivated fetal bovine serum (FBS, Gibco). RNAi treatments were carried out according to ref. (*Somma et al., 2008*). dsRNA-treated cells were grown for 4–5 days at 25°C, and then processed for cytological and biochemical analyses. Drug treatments in S2 cells were all performed at 25°C for 30 min: colchicine 100 µM, RNAse 0,1 mg/ml. HeLa cells (ATCC, RRID: CVCL_0030; tested negative for mycoplasma) were grown in DMEM Glutamax (Invitrogen) supplemented with 10% FBS at 37°C in a humidified incubator with 5% $CO_2$. HeLa cells were transfected with siRNAs to a non-relevant control mRNA (*LacZ*), *SF3A2* mRNA (SASI_Hs01_00025857, Sigma), *PRP31* mRNA (SASI_Hs01_00048860, Sigma) or *HEC1* mRNA (SASI_Hs01_00138654, Sigma) using the HiPerfect reagent (QIAGEN) according to the manufacturer's instructions. HeLa cells were synchronized at the G1/S boundary using a double-thymidine block; they were treated with 2 mM thymidine for 17 hr, and then released to fresh medium for 10 hr followed by second treatment with 2 mM thymidine for 16 hr. HeLa cells were also synchronized by treatment with 100 ng/ml of nocodazole for 16 hr; mitotic cells from these nocodazole-treated cultures were collected by shaking (time, 0 hr). The remaining adherent cells were washed to remove the drug, incubated in fresh medium for 6 hr and then shaken again to collect mitotic cells (time, 6 hr). To induce microtubule disassembly, HeLa cells were incubated in 3 µM nocodazole for 3 hr.

### dsRNA production

PCR products and dsRNAs were synthesized as described in ref. (*Somma et al., 2008*). The primers used in the PCR reactions were 35 nt long and all contained a 5' T7 RNA polymerase binding site (5'-TAATACGACTCACTATAGGGAGG-3') joined to a gene-specific sequence. The sense and anti-sense gene-specific sequences for *Sf3A2, Prp31, Ndc80, zw10, Sf3A1, Sf3A3, Prp3, Prp4 and Prp8* are described in *Somma et al., 2008*). For *Prp6* and *hoip* we used the following sense and antisense sequences: *Prp6*, sense CCCTGTTGCAGC, antisense TCTTTCGCTGCG; *hoip*, sense AGAGGTTG-CATT, antisense TGATCTGCGACT. The entire *Sf3A2-5'UTR* and *Prp31-3'UTR* sequences (see Fly-Base) fused with the T7 RNA polymerase-binding sites, were cloned in pUC57 vector and then used to make dsRNAs. The mock dsRNA used as control was obtained from an EGFP vector (Clontech); sense: AGCTGTTCACCG, antisense TCACGAACTCCA.

### Generation of cell lines expressing tagged proteins and RNAi rescue experiments

The *Sf3A2* and *Prp31* complete coding sequences were cloned in the pAWR or pAWG vectors in frame with the mRFP and eGFP sequences, respectively (vectors are from the Gateway collection, *Drosophila* Genomics Resource Center, Indiana University). The TAP-tagged Ndc80-containing plasmid is a gift of Michael Goldberg (Cornell University). To perform RNAi rescue experiments and Ndc80 overexpression experiments, we generated stable cell lines expressing Sf3A2-mRFP or Prp31-eGFP under the control of the actin promoter, and Ndc80-TAP under the control of the MtnA promoter. To generate these lines, cells were suspended in Schneider's insect medium supplemented with 10% FBS at a concentration of $1 \times 10^6$ cells/ml, and plated in six-well (1 ml) culture dishes. Each culture was inoculated with 1 µg of Sf3A2-mRFP, Prp31-eGFP or Ndc80-TAP and with 20 ng of pCoHygro plasmid (Invitrogen), in the presence of the Effectene transfection reagent (QIAGEN) according to the manufacturer's instructions. For selection of stably transfected cultures, cells were diluted from 1:5 to 1:10 into the appropriate selective medium 72 hr after transfection.

The SF3A2 and the PRP31 ORFs were rendered siRNA resistant by introducing silent mutations (lower case) in the sequences recognized by the relative siRNAs (*SF3A2-CaGAaATtGCaGAaGGgAT;*

*PRP31-CtGTtAAaGAatTaGGgAA*). The mutated ORFs were produced and cloned in pCDH FLAG by Genewiz Company (USA). Transfection with siRNA-resistant plasmids was performed with Lipofectamine 2000 (Invitrogen) after 24 hr siRNA transfection.

## RT-PCR

RNA extraction was performed with the RNeasy Mini Kit (Qiagen, Germany). For RT-PCR, we used 20 ng of RNA to synthesize cDNAs using the Superscript kit (Invitrogen). For cDNA amplification we used the following primers:

*rp49* F: TACAGGCCCAAGATCGTGAA; *rp49* R: ACGTTGTGCACCAGGAACTT;
*Sf3A1* F: AGCGGTCTGAAACTCTCGCA; *Sf3A1* R: AGCTGAAAGCGTCGCCTTGT
*Sf3A3* F: AAACTGATGGTGGACGAGCA; *Sf3A3* R: GAAGGCGGATAGATCAAGGT
*Prp3* F: CAAGAGGACGTTGAGATTCC; *Prp3* R: ATCTGCAGCTGCTTGGCATT
*Prp4* F: GCGAAGAAGACGTCTTAAGG; *Prp4* R: GGCAATCGACTCTTCGTTGT
*Prp6* F: TGAGTGTGCTCGAGCTGTTT; *Prp6* R: TCCAGAATGGATCGAGCCTT
*Prp8 F*: CGGAGAGGACAAAGATCATC; *Prp8 R*: TCCTTTACCTGCGGATTGTC *hoip* F: TGCAATTGCCCTCAGTGAAG; *hoip* R: CGCTCGATCTCCTGCTGAAT

## Proteomic analysis

We performed a Tandem Mass Tag (TMT) quantitative proteomic analysis of control S2 cells, and of cells depleted of Sf3A2, Prp31 or Sf3A1. For all RNAi samples we checked the quality of RNAi and found that the target protein was below 20% of the control level.

Aliquots of 100 µg of each of the four samples were digested with trypsin (2.5 µg trypsin per 100 µg protein; 37°C, overnight), labeled with Tandem Mass Tag (TMT) six plex reagents according to the manufacturer's protocol (Thermo Fisher Scientific, Loughborough, LE11 5RG, UK) and the labeled samples pooled. A 100 ug aliquot of the pooled sample was evaporated to dryness, resuspended in 5% formic acid and then desalted using a SepPak cartridge according to the manufacturer's instructions (Waters, Milford, MA). Eluate from the SepPak cartridge was again evaporated to dryness and resuspended in buffer A (20 mM ammonium hydroxide, pH 10) prior to fractionation by high-pH reversed-phase chromatography using an Ultimate 3000 liquid chromatography system (Thermo Fisher Scientific). In brief, the sample was loaded onto an XBridge BEH C18 Column (130 Å, 3.5 µm, 2.1 mm X 150 mm, Waters, UK) in buffer A and peptides eluted with an increasing gradient of buffer B (20 mM Ammonium Hydroxide in acetonitrile, pH 10) from 0% to 95% over 60 min. The resulting fractions were evaporated to dryness and resuspended in 1% formic acid prior to analysis by nano-LC MSMS using an Orbitrap Fusion Tribrid mass spectrometer (Thermo Scientific).

High-pH RP fractions were further fractionated using an Ultimate 3000 nano-LC system in line with an Orbitrap Fusion Tribrid mass spectrometer (Thermo Scientific). In brief, peptides in 1% (vol/vol) formic acid were injected onto an Acclaim PepMap C18 nano-trap column (Thermo Scientific). After washing with 0.5% (vol/vol) acetonitrile 0.1% (vol/vol) formic acid peptides were resolved on a 250 mm $\times$75 µm Acclaim PepMap C18 reverse phase analytical column (Thermo Scientific) over a 150 min organic gradient, using seven gradient segments (1–6% solvent B over 1 min, 6–15% B over 58 min, 15–32% B over 58 min, 32–40% B over 5 min, 40–90% B over 1 min, held at 90% B for 6 min and then reduced to 1% B over 1 min) with a flow rate of 300 nl min$^{-1}$. Solvent A was 0.1% formic acid and Solvent B was aqueous 80% acetonitrile in 0.1% formic acid. Peptides were ionized by nano-electrospray ionization at 2.0 kV using a stainless steel emitter with an internal diameter of 30 µm (Thermo Scientific) and a capillary temperature of 275°C.

All spectra were acquired using an Orbitrap Fusion Tribrid mass spectrometer controlled by Xcalibur 2.0 software (Thermo Scientific) and operated in data-dependent acquisition mode using an SPS-MS3 workflow. FTMS1 spectra were collected at a resolution of 120,000, with an automatic gain control (AGC) target of 400,000 and a max injection time of 100 ms. Precursors were filtered with an intensity range from 5000 to 1E20, according to charge state (to include charge states 2–6) and with monoisotopic precursor selection. Previously interrogated precursors were excluded using a dynamic window (60s ± 10ppm). The MS2 precursors were isolated with a quadrupole mass filter set to a width of 1.2 m/z. ITMS2 spectra were collected with an AGC target of 10,000, max injection time of 70 ms and CID collision energy of 35%. For FTMS3 analysis, the Orbitrap was operated at 30,000 resolution with an AGC target of 50,000 and a max injection time of 105 ms. Precursors were

fragmented by high energy collision dissociation (HCD) at a normalized collision energy of 55% to ensure maximal TMT reporter ion yield. Synchronous Precursor Selection (SPS) was enabled to include up to 5 MS2 fragment ions in the FTMS3 scan.

The raw data files were processed and quantified using Proteome Discoverer software v2.1 (Thermo Scientific) and searched against the dmel-all-translation-r5.47 database (21323 entries) using the SEQUEST algorithm. Peptide precursor mass tolerance was set at 10 ppm, and MS/MS tolerance was set at 0.6 Da. Search criteria included oxidation of methionine (+15.9949) as a variable modification and carbamidomethylation of cysteine (+57.0214) and the addition of the TMT mass tag (+229.163) to peptide N-termini and lysine as fixed modifications. Searches were performed with full tryptic digestion and a maximum of two missed cleavages was allowed. The reverse database search option was enabled and the data was filtered to satisfy false discovery rate (FDR) of 5% and to exclude Medium and Low confidence IDs. This analysis identified 4483 protein IDs consistent between the samples. The full dataset can be found at www.thewakefieldlab/ms. The abundance scores for the resultant protein IDs from each of the three RNAi samples (Sf3A1, Sf3A2 and Prp31) were normalised against the control sample to create an Abundance ratio score. The mean Abundance ratio score and standard deviation (S.D.) was calculated. Protein IDs with scores more than 1 s.D. from the mean were identified and classed as significantly reduced, in comparison to control. To identify whether any proteins involved in mitosis were significantly and specifically reduced in the Sf3A2 and Prp31 RNAi cell extracts, that could explain the RNAi phenotype, the significantly reduced protein IDs for each condition were interrogated using a Gene Ontology (GO) classifier (GOTermMapper (https://go.princeton.edu/cgi-bin/GOTermMapper)), concentrating on the GO terms 'cell division' (GO:0051301), 'mitotic cell cycle' (GO:0000278) and 'chromosome segregation' (GO:0007059). These protein IDs were compared between conditions, using the Venn Diagram software available at (http://bioinformatics.psb.ugent.be/cgi-bin/liste/Venn/calculate_venn.htpl).

## Antibody generation

To obtain polyclonal antibodies against *Drosophila* Sf3A2 and Prp31, the entire *Sf3A2* coding sequence and the *Prp31* sequence encoding aa 754–1503 were cloned into pET200 vector (Invitrogen), and the recombinant proteins were purified by electro-elution. Rabbit immunization was carried out by Agro-Bio (La Ferté St Aubin, France) according to standard protocols. The antibodies were then affinity purified as described in ref. (*Chase et al., 2001*).

## Fixation and immunostaining

Preparations of S2 mitotic cells were carried out according to *Somma et al., 2008*). For Ndc80 immunostaining, cells were fixed for 10 min in 4% paraformaldheyde; in all other indirect IF experiments, cells were fixed for 7 min in 3.7% formaldheyde. Brain cell preparations were carried out according to (*Bonaccorsi et al., 2000*). In all cases, immunostaining was performed as described in (*Somma et al., 2008*) using the following antibodies, all diluted in PBS: rabbit anti-Sf3A2 (1:200); rabbit anti-Prp31 (1:200); rabbit anti-Ndc80 (1: 100; a gift of M. Goldberg, Cornell University); mouse anti-α tubulin monoclonal DM1A (1:100; Sigma); chicken anti-Cid (1:10000; *Blower and Karpen, 2001*); rabbit anti-Aurora B (1:100; *Giet and Glover, 2001*); rabbit anti-BubR1 (1:200; *Przewloka et al., 2007*); rabbit anti-CENP-C (1:500; *Heeger et al., 2005*); rabbit anti-CycB (1:100; *Lehner and O'Farrell, 1990*); rabbit anti-Mis12 (1:100; *Przewloka et al., 2007*); rabbit anti-Mitch (1:100; *Williams et al., 2007*); rabbit anti-Kmn1 (1:100; *Przewloka et al., 2007*); rabbit anti-Zw10 (1:100; *Williams et al., 1992*); rabbit anti-Spd2 (1: 3,500; *Giansanti et al., 2008*); rabbit anti-anillin (1:5000; *Giansanti et al., 2015*); rabbit anti-Feo (1:100; *Vernì et al., 2004*). These primary antibodies were detected by incubation for 1 hr with FITC-conjugated anti-mouse (1:100, Jackson Laboratories), Cy3-conjugated anti-rabbit (1:300, Life Technologies), or Cy3-conjugated anti-chicken IgGs (1:100, Jackson Laboratories). Slides were mounted in Vectashield with DAPI (Vector) to stain DNA and reduce fluorescence fading.

HeLa cells were fixed in two ways. For immunostaining of spindles with anti-tubulin, anti-SF3A2, anti-PRP31 or anti-HEC1 antibodies cells were fixed with 4% paraformaldheyde, permeabilized with PBT (0,5% TritonX-100) for 10 min and then in PBS + BSA 1%. For CREST and HEC1 staining, tubulin and HEC1 staining and HURP and HEC1 staining, HeLa cells were rinsed in PEM (20 mM Pipes, 10 mM EGTA, 1 mM MgCl$_2$, pH 6.9), treated for 60 s with 0.1% Triton X-100 in PEM to remove soluble

proteins, and fixed for 10 min in 3.7% paraformaldehyde, 30 mM Sucrose in PEM at RT. Immunostaining was performed with the following primary antibodies all diluted in PBS: anti-α tubulin monoclonal DM1A (1:100, Sigma); monoclonal anti-α-tubulin-FITC (1:200, Sigma); mouse anti-HEC1 (1:100, Abcam, 9G3); human anti-CREST (1:100, Antibodies Inc.), rabbit anti-HURP (1:200, Abcam), rabbit anti-PRP31 (1:100, Santa Cruz) and rabbit anti-SF3A2 (1:100, Sigma), which were detected with FITC-conjugated anti-mouse (1:100, Jackson Laboratories), Cy3-conjugated anti-rabbit (1:300, Life Technologies) or Cy3-conjugated anti-human (1:50, Dako).

All images were captured using a CoolSnap HQ CCD camera (Photometrics; Tucson, AZ) connected to a Zeiss Axioplan fluorescence microscope equipped with an HBO 100 W mercury lamp. To estimate the fluorescence intensities of the halos (*Figure 9D*), we used the ImageJ software. For each cell, we measured the fluorescence of four equally sized regions, two within the halo (usually placed at the opposite sides of the cell) and two outside the cell, in proximity of the halo regions chosen for the analysis. We then calculated the ratio between the average fluorescence of the halo and background regions. To estimate the fluorescence intensities of HEC1 in nocodazole-treated cells (*Figure 9E*), we calculated the ratio between the fluorescence of the area encompassing all the HEC1 kinetochore signals and the background signal from a region outside the cell.

## Live-cell imaging of S2 cells

In vivo analysis was performed on mock-treated, *Sf3A2* and *Prp31* RNAi cells espressing Cherry-tubulin and GFP-Histone (*Goshima et al., 2007*) kindly provided by Gohta Goshima (Nagoya University). Images were collected at 2 min intervals; eight fluorescence optical sections were captured at 1 μm Z steps using a calibrated Prior Proscan stepping motor, with an EM-CCD camera (Cascade II, Photometrics) connected to a spinning-disk confocal head (CarvII, Beckton Dickinson) mounted on an inverted microscope (Eclipse TE2000S, Nikon). Images were acquired using the Metamorph software package (Universal Imaging). Movies were made with the Metamorph software; each fluorescence image shown is a maximum-intensity projection of all sections.

## In vivo *Drosophila* embryo imaging and image analysis

1–2 hr old *Drosophila* embryos expressing α-Tubulin-GFP and Histone-H3-RFP (Bloomington Stock Center) were manually dechorionated, aligned in heptane glue on 22 × 50 mm cover slips and covered with a 1:1 solution of Halocarbon oil 27 and 700 (Sigma). Affinity purified anti-Sf3A2 and anti-Prp31 antibodies were exchanged into injection buffer (100 mM HEPES, pH7.4 and 50 mM KCl), concentrated to 2–5 mg/ml, centrifuged at 13,500 g for 20 min and injected into cycle 10–11 embryos using an Eppendorf Inject Man NI two and Femtotips II needles (Eppendorf). As a control, embryos were injected with BSA (Sigma) dissolved into injection buffer at 5 mg/ml. To assess the dynamic localizations of SF3A2 and Prp31, affinity purified antibodies at ~0.5 mg/ml were first labeled with Dylight 550 NHS Ester (ThermoFisher Cat no.62263), following manufacturer's instructions, and as described in *Conduit et al. (2015)*. Imaging was performed on a Visitron Systems Olympus IX81 microscope with a CSO-X1 spinning disk using a UPlanS APO 1.3 NA (Olympus) 60X objective, with 400 or 800 ms exposure per slice, five slices per stack and a constant room temperature of 22°C.

Image processing and analysis was performed on FIJI. Fluorescence loss caused by bleaching was corrected using the Bleach Corrector macro (developed by Kota Miura, EMBL Heidelberg, Germany). Time-lapse movies were generated of maximum intensity projections of time frames with levels adjusted to reduce background fluorescence. Measurement of cell cycle timings was undertaken manually. NEB to initial chromosome alignment was defined as the time taken from the first frame showing Tubulin-GFP influx into the nucleus to the frame where the condensed chromosomes reached a local maximum X:Y ratio. Anaphase onset was defined as the first frame of initial chromosome segregation. For time-lapse movies in which chromosome segregation was absent, anaphase was defined as the first frame in which the metaphase spindle MT organization was observed to alter (i.e. when spindle MTs appeared to be released form centrosomes in preparation of central spindle formation). Measurements were taken from five embryos for each condition. The mean, standard deviation (SD) and standard error of the mean (SEM) were calculated for the datasets and analyzed for confidence using an unpaired t-test. Spindle length (pole-to-pole) comparisons were undertaken by manually measuring a line from the center of each pair of centrosomes. For line graphs, eight

spindles from the BSA-injected, anti-Sf3A2 and anti-Prp31 injected embryos shown in *Figure 3* (*Videos 5*, *6* and *7*) were measured every 8 s from NEB, the mean and SEM calculated and values plotted in Excel (Microsoft). Statistical significance (t-test) was calculated for each time point. To assess the variability of mature spindle length between injection conditions, 50 spindles from at least four embryos per condition were manually measured, as above, approximately 20 s prior to anaphase onset, and box-and-whisker plots generated on-line in ALCULA.

## Western blotting and co-immunoprecipitation (Co-IP)

For immunoblotting of *Drosophila* proteins, S2 cells were washed in cold PBS and homogenized in lysis buffer (50 mM Hepes KOH pH 7.6, 1 mM MgCl2, 1 mM EGTA, 1% Triton X-100, 45 mM NaF, 45 mM β-glycerophosphate, 0.2 mM Na3VO4) in the presence of a cocktail of protease inhibitors (Roche). Cell extracts were pelleted at 15,000 g in an Eppendorf centrifuge for 15 min at 4°C and the supernatants were analyzed by Western blotting according to (*Somma et al., 2002*), using the following antibodies, all diluted in TBS-T (TBS with 0.1% Tween 20): rabbit anti-Prp31 (1:1000), rabbit anti-Sf3A2 (1:1000), rabbit anti-Ndc80 (1:1000), rabbit anti-Giotto (1:5000; (*Giansanti et al., 2006*)), anti-α tubulin monoclonal DM1A (1:1000, Sigma) anti-lamin monoclonal ADL67.10 (1: 250; Hybridoma Bank). These primary antibodies were detected using HRP conjugated anti-mouse and anti-rabbit IgGs and the ECL detection kit (all from GE Healthcare).

For immunoblotting of human proteins, HeLa cells were washed in cold PBS and homogenized in lysis buffer (50 mM Tris-HCl pH 7.5, 150 mM NaCl, 1 mM EDTA, 1% NP-40) in the presence of a cocktail of protease inhibitors (Roche) and centrifuged as above. Western blotting was performed using the following antibodies, all diluted in TBS-T: anti-PRP31 (1:1000, Santa Cruz Biotechnology), anti-SF3A2 (1:1000, Abcam), anti-HEC1 (1:1000, Abcam), and HRP-conjugated anti-actin (1:5000, Santa Cruz Biotechnology).

For co-IP experiments, both *Drosophila* and HeLa cell extracts were incubated for 3 hr at 25°C with protein G sepharose beads (GE Healthcare) conjugated to antibodies against *Drosophila* (Prp31 and Sf3A2) or human (SF3A2, PRP31and HEC1) proteins. Non-specific rabbit and mouse IgGs (Sigma) were used as co-IP negative controls. After three washes with the lysis buffer and a final wash with 20 mM Tris-HCl pH 7.5, the beads were resuspended in Laemmli buffer. To recover immunoprecipitated proteins and to avoid IgG elution in the reciprocal co-immunoprecipitation experiment between SF3A2 and PRP31, the immunoprecipitated fractions bound to the beads were washed with 0.2% SDS (*Antrobus and Borner, 2011*).

All the chemiluminescent blots were imaged with the ChemiDoc MP imager (Bio-Rad). Band intensities were quantified by densitometric analysis with Image Lab software (Bio-Rad).

## GST-pulldown assays

To obtain GST fusion proteins, full-length *Mitch/Spc25*, *Nuf2*, and *Ndc80* cDNAs were cloned in pGEX-6p1 in frame with the GST sequence at N-term and individually transfected into BL21 cells. Bacterially expressed GST fusion proteins were purified by incubating crude lysates with glutathione sepharose 4B (Amersham), as recommended by the manufacturer. To perform GST pulldown with bacterially expressed and purified proteins, full length *Sf3A2* and *Prp31* sequences fused in frame with −6His at N-term were cloned in pET200 and then expressed in the BL21 (DE3) strain. Exponentially growing cells were induced at 37°C for 4 hr by the addition of 1 mM IPTG. Cells were harvested, resuspended and incubated for 1 hr in lysis buffer (400 mM NaCl, 100 mM KCl, 10% glycerol, 0.5% Triton X-100, 10 mM imidazole, 50 mM Phosphate buffer pH 7.8, 0.2% lysozime, Complete protease inhibitors Roche). Lysates were then sonicated for 20 s, incubated for 30 min with 1.5% N-Lauryl Sarcosyl, and centrifuged for 25 min at 15,000 g at 4°C. The supernatant was then incubated for 1 hr with the Ni-NTA His-Bind Resin (Novagen), extensively washed with lysis buffer containing 20 mM imidazole. His-Proteins were then eluted with lysis buffer containing 250 mM imidazole. GST-Mitch (60 nM), GST-Nuf2 (50 nM), GST-Ndc80 (40 nM) or GST (200 nM) alone were then incubated at 4°C for 2 hr with purified 6His-Sf3A2 (20 nM) or 6His-Prp31 (20 nM) in NETN buffer (20 mM Tris-HCl, pH 8, 100 mM NaCl, 1 mM EDTA, 0.5% NP-40). Complexes were collected by centrifugation, washed thrice with NETN buffer and then resuspended in SDS/PAGE loading buffer. 6His-Sf3A2, 6His-Prp31 were detected with anti-His HRP-conjugated (1:500, Roche) or rabbit anti-Prp31 (1:1000), rabbit anti-Sf3A2 (1:1000) antibodies. To perform GST pulldown from extracts,

full-length Sf3A2 and Prp31 cDNAs were cloned into a pAWH vector in frame with 3xHA tag at C-term (Drosophila Genome Resource Center, Bloomington, IN) and transfected into S2 cells using Cellfectin (Invitrogen). Lysates from S2 cells expressing HA-tagged SFs (prepared as above) were incubated at 4°C for 2 hr with GST-fusion proteins or GST alone and analyzed by western blotting using mouse monoclonal anti-HA (1:1000; Roche).

## MT co-sedimentation assays

Bacterially expressed 6His-Prp31and 6His-Sf3A2 were purified under mild, non-denaturing conditions. 6His-tagged proteins were eluted using a lysis buffer containing 300 mM imidazole. All proteins were concentrated with Centricon Millipore and resuspended in 80 mM PIPES containing 25 mM NaCl. The MT binding assay was performed using the Microtubule Binding Protein Spin-Down Assay Biochem Kit from Cytoskeleton (cat. n. BK029). 6His-Prp31 (10 nM) and 6His-Sf3A2 (13 nM) were incubated at RT for 30 min with increasing amounts of stable MTs (obtained according to the manufacturer's instructions), or with 2 µM tubulin in General Tubulin Buffer without Taxol. The mixture was then centrifuged at 100,000 g at RT for 40 min. Supernatants and pellets were collected and analyzed by Coomassie staining or by western blotting. The fraction of proteins bound to microtubules was plotted as a function of MT concentration.

## Acknowledgements

We thank David Glover, Mike Goldberg, Gohta Goshima, Gary Karpen, Christian Lehner and Stefano Sechi for their generous gifts of antibodies and cell lines, and Giuseppe Bosso for the GST-tagged telomere proteins. We thank Kate Heesom, of the Bristol Proteomics Facility, for processing the TMT quantitative proteomics samples, and Ryan Ames for assisting in the proteome analysis. The monoclonal anti-lamin Dm0 was obtained from the Hybridoma Bank, created by the NICHD of the NIH and maintained at the University of Iowa, Department of Biology, Iowa City, IA 52242. This work was supported by grants from Italian Association for Cancer Research to MG (AIRC, IG16020 and 20528), a PRIN grant from MIUR to SB, a BBSRC grant (BB/K017837/1) to JGW and a BBSRC PhD studentship to DH.

## Additional information

### Funding

| Funder | Grant reference number | Author |
|---|---|---|
| Associazione Italiana per la Ricerca sul Cancro | IG16020 | Maurizio Gatti |
| Ministero dell'Istruzione, dell'Università e della Ricerca | | Silvia Bonaccorsi |
| Biotechnology and Biological Sciences Research Council | BB/K017837/1 | James G Wakefield |
| Associazione Italiana per la Ricerca sul Cancro | IG20528 | Maurizio Gatti |

The funders had no role in study design, data collection and interpretation, or the decision to submit the work for publication.

### Author contributions

Claudia Pellacani, Conceptualization, Validation, Investigation, Methodology; Elisabetta Bucciarelli, Data curation, Formal analysis, Investigation, Visualization; Fioranna Renda, Data curation, Formal analysis, Investigation; Daniel Hayward, Jack Chen, Investigation, Visualization; Antonella Palena, Investigation, Methodology; Silvia Bonaccorsi, Data curation, Formal analysis, Funding acquisition; James G Wakefield, Conceptualization, Formal analysis, Funding acquisition, Investigation; Maurizio Gatti, Conceptualization, Supervision, Funding acquisition, Writing—original draft, Writing—review and editing; Maria Patrizia Somma, Conceptualization, Data curation, Investigation, Writing—original draft, Writing—review and editing

## Author ORCIDs

Maurizio Gatti http://orcid.org/0000-0003-3777-300X
Maria Patrizia Somma http://orcid.org/0000-0002-7585-3484

## Decision letter and Author response

Decision letter https://doi.org/10.7554/eLife.40325.045
Author response https://doi.org/10.7554/eLife.40325.046

## Additional files

### Supplementary files

• Supplementary file 1. Lists of proteins whose abundance is significantly reduced (1 SD) in *Prp31*, *Sf3A2* or *Sf3A1* RNAi cells, normalized against control. Proteins with Gene Ontologies (GOs) of mitotic cell cycle, cell division or chromosome segregation are highlighted in grey.
DOI: https://doi.org/10.7554/eLife.40325.043

• Transparent reporting form
DOI: https://doi.org/10.7554/eLife.40325.044

### Data availability

All data generated or analyzed during this study are included in the manuscript and Supplementary File 1. The full dataset for proteomic analyses reported in Figures 6 and Supplementary File 1 can be found at https://www.thewakefieldlab.com/ms; the significantly reduced protein IDs for each RNAi experiment were interrogated using a Gene Ontology (GO) classifier (GOTermMapper (https://go.princeton.edu/cgi-bin/GOTermMapper), concentrating on the GO terms "cell division" (GO: 0051301), "mitotic cell cycle" (GO:0000278) and "chromosome segregation" (GO:0007059). Source data files have been provided for Figures 1, 4, 5, 7, 8, and 9.

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
