## [Decision Letter]

[Editors’ note: a previous version of this study was rejected after peer review, but the authors submitted for reconsideration. The first decision letter after peer review is shown below.]

Thank you for submitting your work entitled "Splicing Factors Sf3A2 and Prp31 have direct roles in mitotic chromosome segregation" for consideration by *eLife*. Your article has been reviewed by a Senior Editor, a Reviewing Editor, and three reviewers. The following individuals involved in review of your submission have agreed to reveal their identity: Helder Maiato (Reviewer #2); William Sullivan (Reviewer #3).

Our decision has been reached after consultation between the reviewers. Based on these discussions and the individual reviews below, we regret to inform you that in its present form your work will not be considered further for publication in *eLife*. The major concern of all the reviewers was whether the effect of inhibiting Sf3A2 and Prp31 impacts directly on mitosis or is indirect, which is the major point of interest of the paper. You will see from the attached reviews that a number of control experiments are suggested; the injection of antibodies in mitosis would be a particularly important in this context. In our opinion these experiments will take longer than the period allowed for resubmission therefore we are rejecting your manuscript. Nevertheless, demonstrating that splicing factors have a direct role in mitosis would be an interesting finding; therefore, should you generate data that allow more definitive conclusions to be reached we would be prepared to review a new submission.

Reviewer #2:

The manuscript by Gatti et al., investigate a direct role for the conserved splicing factors Sf3A2 and Prp31 in mitosis. Several splicing factors have previously been implicated in mitosis, notoriously after RNAi perturbation, which could argue against a direct role. In the present paper, the authors report similar findings after RNAi-mediated depletion of the Sf3A2 and Prp31 (and respective orthologues) in *Drosophila* S2 cells in culture, larval neuroblasts and human HeLa cells. They go further do demonstrate that antibody injections in *Drosophila* syncytial embryos cause problems in mitosis within approximately 1 minute after injection, arguing for a specific role in mitosis. The phenotypes in the three different *Drosophila* models are nevertheless quite distinct. Phenotypes after RNAi in HeLa cells are more related with those observed in S2 cells. The authors provide compelling evidence that Sf3A2 and Prp31 bind to each other and to Ndc80, likely in a direct fashion. Sf3A2 and Prp31 also appear to associate with microtubules and localize in the spindle region during mitosis (both in S2 and HeLa cells). Sf3A2 and Prp31 appear to be required for normal recruitment of Ndc80 to kinetochores also in both systems. Overall, the authors provide a strong case for a direct role of Sf3A2 and Prp31 in regulating Ndc80 recruitment to kinetochores during mitosis. There are however some major concerns that would merit some attention by the authors. This is particularly important because this field is quite murky and any major claim for a splicing-unrelated role of splicing factors in mitosis must be bullet proof.

1) The authors focused their functional analysis of Sf3A2 and Prp31 on the impact over Ndc80. The main problem here is that one cannot be sure what else is being affected, especially given some phenotypic differences between Ndc80 and Sf3A2 and Prp31. The authors made an effort to look at the localization of other kinetochore components, but this falls short to draw any definitive conclusion. A quantitative proteomic analysis would be ideal here comparing controls with Sf3A2 and Prp31 RNAis, but this would require a significant additional effort. Could the authors show at least a Silver-stained gel comparing total protein amounts between controls and Sf3A2 and Prp31 RNAis? If major differences are found, would be important to identify which proteins are affected (by mass-spec, but now against specific proteins).

2) Along the same line, but now focusing on Ndc80, could the expression of an Ndc80 cDNA (which is already spliced) rescue at least part (the PMLES) of the Sf3A2 and Prp31 phenotypes? This experiment might turn out to be informative on multiple fronts, as it will test whether the observed mitotic phenotypes after Sf3A2 and Prp31 are dependent or not on Ndc80 splicing. It will also be important to rescue the mitotic phenotypes after Sf3A2 and Prp31 RNAis, by expressing the corresponding ORFs. This can be done by targeting the 5' or 3' UTR region of the endogenous genes, making sure these are not present in the rescue constructs (see e.g. Lince-Faria et al., 2009)

3) The other matter of concern relates with the phenotypic differences between RNAis and antibody microinjections. In the latter, the main defect appears to be after congression, arguing for a kinetochore-independent role for Sf3A2 and Prp31. Overall, it is difficult to reconcile all the data to draw a strong conclusion here. It should be noted that most injections were performed in interphase nuclei, leaving open the possibility that splicing might still be affected (there are still few short transcripts that are spliced between cycles 8-13). In the same line, the total duration of interphase at these stages is around 5 minutes, and splicing events can take as short as 30 seconds (see e.g. Guilgur et al., 2014, cited by the authors). One suggestion would be to extend the few examples where embryos were injected apparently (no indication of injection time is provided) already in mitosis. An easier way might be to perform the injections in metaphase-arrested embryos by inhibiting the APC (see e.g. Oliveira et al., 2010).

4) The spindle MT localization of Sf3A2 and Prp31 is not totally convincing. Many proteins, including soluble tubulin, are known to enrich in the spindle region, simply because of molecular crowding due to a fenestrated envelope and the presence of large organelles that are excluded from the spindle region (see e.g. Schweizer et al., 2013). This turns out to be MT-independent. The authors do show one colchicine-treated cell in figure S7C, but there is no information in the Materials and methods section about this experiment, including the drug concentration used. I would suggest to provide a better characterization of Sf3A2 and Prp31 localization in mitosis after 30 minutes treatment with 100 μM Colchicine, which is sufficient to depolymerize all spindle MTs in S2 cells. Longer treatments (for 2 hours as used by the authors) will cause diffusion of "spindle matrix" components due to gradual disintegration of the membranous envelope that surrounds the spindle. It would also be important to provide direct localization of Sf3A2 and Prp31 in mitosis by expressing a GFP-tagged version. Similar approaches should also be conducted in human cells with micromolar doses of nocodazole.

5) The time course experiment after double thymidine block is suggestive of a specific mitotic interaction between Sf3A2, Prp31 and HEC1. However, the possibility remains that many of the cells after 8 hours release are actually in G2. This experiment should be repeated using mitotic extracts only, obtained by mitotic shake-off of cells treated with nocodazole or STLC. The kinetochore localization of HEC1 after Sf3A2 and Prp31 RNAi in HeLa cells should also be investigated in cells treated with nocodazole (to standardize the conditions) and properly quantified relative to constitutive kinetochore components (e.g. using ACA serum from CREST patients).

Reviewer #3:

The manuscript by Pellacani et al., examines the role of splicing factors mitotic chromosome segregation. Specifically, the authors address an unresolved long-standing issue of whether the mitotic defects associated with mutants in splicing factors are an indirect consequence of a failure to properly splice factors required for mitosis or whether the splicing factors have a distinct direct role in mitosis. The studies presented here provide a comprehensive set of experiments favoring the latter. Key evidence includes inhibitory antibody injections of splicing factors that result in an immediate disruption of chromosome segregation, and localization studies demonstrating these splicing factors localize to spindle microtubules and the Ndc80. Significantly they also demonstrate that these factors are required for kinetochore localization of Ndc80.

Overall the manuscript provides a thorough and compelling set of experiments that strongly support a direct action of splicing factors on the mitotic machinery. The work is of high quality and for the most part the conclusions are well supported by the data. Once the issue below is addressed this manuscript is well suited for publication in *eLife*:

1) The authors find that their RNAi/cell and antibody/embryo experiments produce different phenotypes. One key difference between the two experiments is that the former results in knockdown of the splicing factors throughout the cell cycle while the latter only during entry into metaphase.

In addition, some of the phenotypes observed are consistent with inhibition of S-phase (see aphidicolin embryo injection movies in Fasulo et al., 2011). To determine if there are interphase effects of splicing factor disruption the authors should inject the antibody during late anaphase and film the following cycle. If S-phase is inhibited there will be a dramatic delay in interphase. In addition, there will be chromosome segregation defects similar to those they observe.

Reviewer #4:

This manuscript describes the roles of two conserved splicing factors in mitosis in *Drosophila* and humans. Defective phenotypes after RNAi are well described. I may have missed something, but experiments which exclude the possibility of off-target effects appear to be missing. If these control experiments have been done, they should be clearly stated in the main text with figures. They are essential before any publication, as many wrong papers have been published in the past due to off-target effects.

The authors conclude that this mitotic role is independent of splicing function. This conclusion is based on observation of the mitotic defect within 1 minute after antibody injection. This evidence is convincing in principle, but I have a reservation on the strength of the control they used (see below). As this is a crucial experiment for this study, it is important to use a more robust control.

Apart from phenotypic analysis, some mechanistic studies have been carried out. They showed a reduction of the Ndc80 complex on kinetochores, and interaction between Ndc80 and these splicing factors, which explain the phenotype they observed. It is still not clear how these proteins promote the recruitment of the Ndc80 complex to kinetochores. I judge that this is a very good but not in-depth mechanistic study. Therefore, provided the above two experiments are done, a merit of publishing this study in *eLife* depends on how significant direct involvement of splicing factors in mitosis is judged to be.

Many examples of proteins with multiple unrelated roles (moonlighting) have been documented. In the old days, it was often proposed that they have crucial roles in cross-talk between two processes, but these claims have not been substantiated in most cases. I do not think that splicing factors having direct mitotic functions conceptually advance the understanding of either splicing or mitosis. Therefore, when considering the merit of this study for publication in *eLife*, it should be treated in a similar way to the discovery of a novel conserved proteins which is important for mitosis.

1) Control experiments are required to exclude the possibility of off-target effects of RNAi in *Drosophila* or humans. Ideally rescue experiments should be carried out using an RNAi resistant wild-type gene.

2) Figure 3/subsection “The phenotypes elicited by Sf3A2 or Prp31 inhibition in the embryo suggest a direct mitotic role of these SFs”. Injection of buffer only was used as a control. This is acceptable in some cases, but as this is a critical experiment assessed by a sensitive assay, a further control is essential. Preparing affinity purified antibodies requires multiple chemicals which may affect cell physiology even in a trace amount upon injection into cells. Therefore, unrelated antibodies prepared in the identical way (ideally at the same time) should be used as a control. As the authors' claim for direct roles heavily relies on this particular experiment, it is critical to do this control carefully.

[Editors’ note: what now follows is the decision letter after the authors submitted for further consideration.]

Thank you for submitting your article "Splicing Factors Sf3A2 and Prp31 have direct roles in mitotic chromosome segregation" for consideration by *eLife*. Your article has been reviewed by four peer reviewers, including Jon Pines as the Reviewing Editor and Reviewer #1, and the evaluation has been overseen by a Reviewing Editor and Anna Akhmanova as the Senior Editor. The following individuals involved in review of your submission have agreed to reveal their identity: Helder Maiato (Reviewer #2); William Sullivan (Reviewer #3).

The reviewers have discussed the reviews with one another and the Reviewing Editor has drafted this decision to help you prepare a revised submission.

Summary:

The manuscript by Gatti et al., addresses whether the splicing factors Sf3A2 and Prp31 have direct mitotic roles, independently from splicing. In this revised manuscript the authors have made considerable efforts to address previous criticisms of their study. It seems to me that they have done pretty much as much as they can to exclude the possibility that the splicing factors are acting indirectly – their analysis of transcripts is particularly important here. The biochemical data showing interaction between these two factors with MTs and Ndc80 offer a plausible explanation for some of the observed phenotypes and suggest that mitosis is immune to potential splicing defects in the several mRNAs that are under control of Sf3A2 and Prp31.

Essential revisions:

Before publication the authors should address several inconsistencies in the data.

1) Ndc80 localization at kinetochores seems reduced after perturbation of Sf3A2 and Prp31only in the presence of MTs. Ndc80/Hec1 localization is normal without MTs (at least in HeLa cells). Thus, SFs are not required for Ndc80 recruitment but might be involved in its maintenance at KTs. Consistently, the in vivo interaction between Sf3A2 and Prp31 with Ndc80 does not seem to take place in mitotic cells in the absence of MTs, leading the authors to propose a model in which the Ndc80/Hec1-SF interaction requires the presence of spindle MTs. The problem comes from the GST pull-down experiments that clearly show that purified recombinant Ndc80 and SFs all interact, and this happens in the absence of MTs. Together with the fact that the observed mitotic phenotypes related with the perturbation of SFs (e.g. monopolar spindles) are not consistent with loss of Ndc80, leaves open an obvious gap in the model. This cannot simply be explained by differences in SAC activity among different cells/systems. These points should be addressed either by modifying the model to include caveats and alternative interpretations – notably that there may be other targets than Ndc80 – or by including experimental data to support the proposed model.

2) The immunofluorescence data remains somewhat difficult to interpret. The authors state that detergent pre-extraction before fixation improves visualization of KT proteins, but this is also known to cause artefacts. The reviewers do not see the need for such harsh conditions and are aware that conventional PFA works well to stain Hec1; even methanol fixation is preferable to pre-extraction. The authors should include data using a different staining technique that avoids pre-extraction.

---

## [Author Response]

[Editors’ note: the author responses to the first round of peer review follow.]

Our decision has been reached after consultation between the reviewers. Based on these discussions and the individual reviews below, we regret to inform you that in its present form your work will not be considered further for publication in eLife. The major concern of all the reviewers was whether the effect of inhibiting Sf3A2 and Prp31 impacts directly on mitosis or is indirect, which is the major point of interest of the paper. You will see from the attached reviews that a number of control experiments are suggested; the injection of antibodies in mitosis would be a particularly important in this context. In our opinion these experiments will take longer than the period allowed for resubmission therefore we are rejecting your manuscript. Nevertheless, demonstrating that splicing factors have a direct role in mitosis would be an interesting finding; therefore, should you generate data that allow more definitive conclusions to be reached we would be prepared to review a new submission.Reviewer #2:The manuscript by Gatti et al., investigate a direct role for the conserved splicing factors Sf3A2 and Prp31 in mitosis. Several splicing factors have previously been implicated in mitosis, notoriously after RNAi perturbation, which could argue against a direct role. In the present paper, the authors report similar findings after RNAi-mediated depletion of the Sf3A2 and Prp31 (and respective orthologues) in Drosophila S2 cells in culture, larval neuroblasts and human HeLa cells. They go further do demonstrate that antibody injections in Drosophila syncytial embryos cause problems in mitosis within approximately 1 minute after injection, arguing for a specific role in mitosis. The phenotypes in the three different Drosophila models are nevertheless quite distinct. Phenotypes after RNAi in HeLa cells are more related with those observed in S2 cells. The authors provide compelling evidence that Sf3A2 and Prp31 bind to each other and to Ndc80, likely in a direct fashion. Sf3A2 and Prp31 also appear to associate with microtubules and localize in the spindle region during mitosis (both in S2 and HeLa cells). Sf3A2 and Prp31 appear to be required for normal recruitment of Ndc80 to kinetochores also in both systems. Overall, the authors provide a strong case for a direct role of Sf3A2 and Prp31 in regulating Ndc80 recruitment to kinetochores during mitosis. There are however some major concerns that would merit some attention by the authors. This is particularly important because this field is quite murky and any major claim for a splicing-unrelated role of splicing factors in mitosis must be bullet proof.1) The authors focused their functional analysis of Sf3A2 and Prp31 on the impact over Ndc80. The main problem here is that one cannot be sure what else is being affected, especially given some phenotypic differences between Ndc80 and Sf3A2 and Prp31. The authors made an effort to look at the localization of other kinetochore components, but this falls short to draw any definitive conclusion. A quantitative proteomic analysis would be ideal here comparing controls with Sf3A2 and Prp31 RNAis, but this would require a significant additional effort. Could the authors show at least a Silver-stained gel comparing total protein amounts between controls and Sf3A2 and Prp31 RNAis? If major differences are found, would be important to identify which proteins are affected (by mass-spec, but now against specific proteins).

We performed Tandem Mass Tag (TMT) quantitative proteomic analysis of control S2 cells and of cells depleted of Sf3A2, Prp31 or Sf3A1; Sf3A1 is a splicing factor that does not exhibit any apparent mitotic defect upon its RNAi-mediated depletion. This analysis (Figure 6 and supplementary file of Figure 6) showed that 99, 136 and 948 proteins were significantly reduced compared to control following *Sf3A2, Prp31* or *Sf3A1* RNAi, respectively. A Gene Ontology (GO) classifier (GOTermMapper (https://go.princeton.edu/cgi-bin/GOTermMapper)), concentrating on the GO terms “cell division”, “mitotic cell cycle” and “chromosome segregation identified 9, 12 and 50 IDs for *Sf3A2, Prp31* and *Sf3A1* RNAi cells, respectively (Figure 6 and supplementary file of Figure 6). Only a single protein, HUS1-like, was significantly reduced in both *Prp31* and *Sf3A2*, but not *Sf3A1*, RNAi cells. As HUS1-like is involved in homologous recombination repair during meiosis (Peretz et al., 2009), it is unlikely to be the causative factor of the mitotic phenotype observed in SF-depleted S2 cells.

2) Along the same line, but now focusing on Ndc80, could the expression of an Ndc80 cDNA (which is already spliced) rescue at least part (the PMLES) of the Sf3A2 and Prp31 phenotypes? This experiment might turn out to be informative on multiple fronts, as it will test whether the observed mitotic phenotypes after Sf3A2 and Prp31 are dependent or not on Ndc80 splicing. It will also be important to rescue the mitotic phenotypes after Sf3A2 and Prp31 RNAis, by expressing the corresponding ORFs. This can be done by targeting the 5' or 3' UTR region of the endogenous genes, making sure these are not present in the rescue constructs (see e.g. Lince-Faria et al., 2009)

We have now quantitated Ndc80 in SF-depleted cells and shown that its amount is not reduced compared to mock controls (Figure 7D). Furthermore, Ndc80 was not significantly reduced in our proteomic analyses of RNAi cells (Supplementary file of Figure 6). We have also performed RNAi against *Sf3A2* and *Prp31* in an S2 cell line bearing a construct containing Tap-tagged *Ndc80* cDNA under the control of the inducible Mtn promoter. Expression of Ndc80-Tap, which localized normally to the kinetochores, did not rescue the mitotic phenotype elicited by the SF depletion (Figure 8). Together, these results indicate that mitotic defects observed in Sf3A2- and Prp31-depleted cellsare not caused by Ndc80 deficiency.

We performed the suggested rescue experiments. We found that dsRNAs targeting the 5' UTR of *Sf3A2* or the 3' UTR of *Prp31* produce the same mitotic phenotypes as those produced by the dsRNAs directed to the coding sequences; these phenotypes were rescued by expression of RNAi resistant genes (*Sf3A2-RFP* and *Prp31-GFP*) lacking the UTRs (Figure 1D and Figure 1—figure supplement 3), ruling out the possibility of RNAi off-target effects.

3) The other matter of concern relates with the phenotypic differences between RNAis and antibody microinjections. In the latter, the main defect appears to be after congression, arguing for a kinetochore-independent role for Sf3A2 and Prp31. Overall, it is difficult to reconcile all the data to draw a strong conclusion here. It should be noted that most injections were performed in interphase nuclei, leaving open the possibility that splicing might still be affected (there are still few short transcripts that are spliced between cycles 8-13). In the same line, the total duration of interphase at these stages is around 5 minutes, and splicing events can take as short as 30 secseconds (see e.g. Guilgur et al., 2014, cited by the authors). One suggestion would be to extend the few examples where embryos were injected apparently (no indication of injection time is provided) already in mitosis. An easier way might be to perform the injections in metaphase-arrested embryos by inhibiting the APC (see e.g. Oliveira et al., 2010).

A qualitative observation suggests that the chromosomes of α-Sf3A2- and α-Prp31-injected embryos are less aligned than in controls throughout the extended metaphase period (Figure 4A, Video 4, Video 5 and Video 6). To test this quantitatively, we measured the area encompassed by the chromosomes during the last 80 s (20 frames) that precede anaphase in anti-SF injected embryos. The chromosomes were, indeed, less tightly aligned than in their mock-injected counterparts (Figure 4E).

We performed additional antibody injection experiments, as suggested (see the embryo injection part of the Results section that is completely rewritten). The experiments confirmed the phenotypes described in the original version of the manuscript. But the crucial piece of information that rules out the possibility of an indirect splicing-related effect of α-Sf3A2- and α-Prp31 is the fact the fluorescently-labeled antibodies are virtually excluded from interphase and prophase nuclei; they penetrate into the nuclear space only at prometaphase when the nuclear envelope becomes fenestrated (Figure 4—figure supplement 3). This finding, which we did not emphasize in the original version of the manuscript, strongly suggests that during interphase the antibodies cannot disrupt the splicing process that occurs within the nucleus. Also, even if the antibodies are injected one minute before nuclear fenestration, they can act on the mitotic apparatus only after their entry into the nuclear space at prometaphase. Thus, given that splicing does not occur during mitosis, our embryo injection experiments make it highly unlikely an indirect, splicing-dependent mitotic effect of Sf3A3 and Prp31 inhibition.

An additional consideration that makes it extremely unlikely that antibody-induced SF inactivation disrupts mitosis by affecting the splicing of a mitotic mRNA is that the protein encoded by this hypothetical mRNA would be subject to an extraordinary rapid turnover, so that it would need to be continuously replenished.

4) The spindle MT localization of Sf3A2 and Prp31 is not totally convincing. Many proteins, including soluble tubulin, are known to enrich in the spindle region, simply because of molecular crowding due to a fenestrated envelope and the presence of large organelles that are excluded from the spindle region (see e.g. Schweizer et al., 2013). This turns out to be MT-independent. The authors do show one colchicine-treated cell in figure S7C, but there is no information in the Materials and methods section about this experiment, including the drug concentration used. I would suggest to provide a better characterization of Sf3A2 and Prp31 localization in mitosis after 30 minutes treatment with 100 μM Colchicine, which is sufficient to depolymerize all spindle MTs in S2 cells. Longer treatments (for 2 hours as used by the authors) will cause diffusion of "spindle matrix" components due to gradual disintegration of the membranous envelope that surrounds the spindle. It would also be important to provide direct localization of Sf3A2 and Prp31 in mitosis by expressing a GFP-tagged version. Similar approaches should also be conducted in human cells with micromolar doses of nocodazole.

We followed the reviewer’s suggestion. We incubated the cells for 30 minutes in 100 μM colchicine, a treatment that is sufficient to depolymerize the spindle MTs in S2 cells but insufficient to cause diffusion of the spindle matrix components (Schweizer et al., 2013). We found that after this treatment both Sf3A2 and Prp31 are diffuse and do not show any enrichment in any spindle-like structure (i. e. the spindle matrix) (Figure 5—figure supplement 2A, B). We performed analogous experiments in HeLa cells; brief nocodazole treatments led to a diffuse localization of the SFs in metaphase cells, suggesting they associate with the spindle and not with the spindle matrix (Figure 9A).

We also examined S2 cells expressing of *Sf3A2-RFP* or *Prp31-GFP* under the control of the actin promoter and found that the tagged proteins accumulate on the metaphase spindles (Figure 5—figure supplement 1C).

5) The time course experiment after double thymidine block is suggestive of a specific mitotic interaction between Sf3A2, Prp31 and HEC1. However, the possibility remains that many of the cells after 8 hours release are actually in G2. This experiment should be repeated using mitotic extracts only, obtained by mitotic shake-off of cells treated with nocodazole or STLC. The kinetochore localization of HEC1 after Sf3A2 and Prp31 RNAi in HeLa cells should also be investigated in cells treated with nocodazole (to standardize the conditions) and properly quantified relative to constitutive kinetochore components (e.g. using ACA serum from CREST patients).

We performed the suggested experiment. We synchronized HeLa cells with a 16 h nocodazole treatment; mitotic cells were then collected by mechanical shake-off both at the end of nocodazole treatment and 6 hours after the drug removal. As expected, both cell populations were highly enriched in cells undergoing mitosis marked by the PHH3, which was not found in adherent cells (Figure 10B). Co-IP with an anti-HEC1 antibody did not detect any HEC1-SF interaction in cells shaken-off from nocodazole-containing cultures or from adherent cells. However, mitotic cells shaken off from cultures kept in fresh medium for 6 hours after nocodazole removal displayed a robust interaction between HEC1 and either SF3A2 or PRP31. This is a very interesting finding, as it indicates that HEC1-SF interaction does not occur in mitotic cells with depolymerized MTs, but takes place only when these cells have reassembled a spindle, suggesting that the HEC1-SF interaction requires the presence of the spindle MTs.

We have also investigated kinetochore localization of HEC1 in *SF3A2* and *PRP31* RNAi HeLa cells treated with nocodazole. In RNAi cells treated with nocodazole (and fixed after pre-extraction with Triton X-100) the HEC1 signals were compact and strictly associated with the kinetochores (Figure 11—figure supplement 1). In contrast, in RNAi cells not exposed to nocodazole (also fixed after pre-extraction with Triton X-100), HEC1 was associated with filaments that are stained by anti-tubulin antibodies and are likely to be kinetochore fibers. These observations strongly suggest that SF3A2 and PRP31 are required to anchor HEC1 to the kinetochores and that, in the absence of these SFs, HEC1 tends to diffuse along the kinetochore MTs.

Reviewer #3:The manuscript by Pellacani et al., examines the role of splicing factors mitotic chromosome segregation. Specifically, the authors address an unresolved long-standing issue of whether the mitotic defects associated with mutants in splicing factors are an indirect consequence of a failure to properly splice factors required for mitosis or whether the splicing factors have a distinct direct role in mitosis. The studies presented here provide a comprehensive set of experiments favoring the latter. Key evidence includes inhibitory antibody injections of splicing factors that result in an immediate disruption of chromosome segregation, and localization studies demonstrating these splicing factors localize to spindle microtubules and the Ndc80. Significantly they also demonstrate that these factors are required for kinetochore localization of Ndc80.Overall the manuscript provides a thorough and compelling set of experiments that strongly support a direct action of splicing factors on the mitotic machinery. The work is of high quality and for the most part the conclusions are well supported by the data. Once the issue below is addressed this manuscript is well suited for publication in eLife:1) The authors find that their RNAi/cell and antibody/embryo experiments produce different phenotypes. One key difference between the two experiments is that the former results in knockdown of the splicing factors throughout the cell cycle while the latter only during entry into metaphase.In addition, some of the phenotypes observed are consistent with inhibition of S-phase (see aphidicolin embryo injection movies in Fasulo et al., 2011). To determine if there are interphase effects of splicing factor disruption the authors should inject the antibody during late anaphase and film the following cycle. If S-phase is inhibited there will be a dramatic delay in interphase. In addition, there will be chromosome segregation defects similar to those they observe.

We performed the experiment suggested by the reviewer and injected embryos with either of the SF antibodies during late anaphase/telophase. The embryos did not exhibit an increase in the length of the following S phase compared to the mock-injected control embryos, suggesting that interphase/S phase was not disrupted. However, when the embryos entered mitosis they displayed the same defects seen in embryos injected just before or during mitosis (Figure 4—figure supplement 1). We note that it is quite possible that the antibodies injected in late anaphase/telophase did not enter the nuclei due to the reassembly of the nuclear envelope. In this case, the antibodies might have been excluded from the nucleus till next mitosis. Indeed, we have shown that the antibodies are virtually excluded from interphase and prophase nuclei and penetrate into the nuclear space only when the nucleus becomes fenestrated in prometaphase (Figure 4—figure supplement 3). This is an important observation that (unfortunately) we did not emphasize in the original version of our manuscript. This finding strongly suggests that the antibodies cannot disrupt the splicing process during interphase, and even if they are injected one minute before mitosis, they can act on kinetochores/microtubules only after their entry into the nucleus at prometaphase. Thus, given that splicing does not occur during mitosis, this finding makes highly unlikely an indirect, splicing-dependent mitotic effect of Sf3A3 and Prp31 inhibition.

Reviewer #4:This manuscript describes the roles of two conserved splicing factors in mitosis in Drosophila and humans. Defective phenotypes after RNAi are well described. I may have missed something, but experiments which exclude the possibility of off-target effects appear to be missing. If these control experiments have been done, they should be clearly stated in the main text with figures. They are essential before any publication, as many wrong papers have been published in the past due to off-target effects.The authors conclude that this mitotic role is independent of splicing function. This conclusion is based on observation of the mitotic defect within 1 minute after antibody injection. This evidence is convincing in principle, but I have a reservation on the strength of the control they used (see below). As this is a crucial experiment for this study, it is important to use a more robust control.Apart from phenotypic analysis, some mechanistic studies have been carried out. They showed a reduction of the Ndc80 complex on kinetochores, and interaction between Ndc80 and these splicing factors, which explain the phenotype they observed. It is still not clear how these proteins promote the recruitment of the Ndc80 complex to kinetochores. I judge that this is a very good but not in-depth mechanistic study. Therefore, provided the above two experiments are done, a merit of publishing this study in eLife depends on how significant direct involvement of splicing factors in mitosis is judged to be.Many examples of proteins with multiple unrelated roles (moonlighting) have been documented. In the old days, it was often proposed that they have crucial roles in cross-talk between two processes, but these claims have not been substantiated in most cases. I do not think that splicing factors having direct mitotic functions conceptually advance the understanding of either splicing or mitosis. Therefore, when considering the merit of this study for publication in eLife, it should be treated in a similar way to the discovery of a novel conserved proteins which is important for mitosis.1) Control experiments are required to exclude the possibility of off-target effects of RNAi in Drosophila or humans. Ideally rescue experiments should be carried out using an RNAi resistant wild-type gene.

We performed rescue experiments for both *Drosophila* and human cells. dsRNAs targeting the 5' UTR of *Sf3A2* or the 3' UTR of *Prp31* produced the same mitotic phenotypes as the dsRNAs directed to the coding sequences; these phenotypes were rescued by expression of RNAi resistant genes (*Sf3A2-RFP* and *Prp31-GFP*) lacking the UTRs (Figure 1D and Figure 1—figure supplement 3), ruling out the possibility off-target effects of RNAi. The HeLA cell phenotypes caused by *SF3A2* and *PRP31* silencingwith specific siRNAs were rescued by the ectopic expression of FLAG-tagged *SF3A2* or *PRP31* genes bearing translationally silent mutations that render them insensitive to these siRNAs (Figure 9C).

2) Figure 3/subsection “The phenotypes elicited by Sf3A2 or Prp31 inhibition in the embryo suggest a direct mitotic role of these SFs”. Injection of buffer only was used as a control. This is acceptable in some cases, but as this is a critical experiment assessed by a sensitive assay, a further control is essential. Preparing affinity purified antibodies requires multiple chemicals which may affect cell physiology even in a trace amount upon injection into cells. Therefore, unrelated antibodies prepared in the identical way (ideally at the same time) should be used as a control. As the authors' claim for direct roles heavily relies on this particular experiment, it is critical to do this control carefully.

We carried out several additional injection experiments using new aliquots of affinity-purified antibodies. These experiments gave results fully comparable to those described in the original version of this manuscript (see the Results and Figure 4—figure supplement 1 as an example of the new results). To rule out the possibility that the specific phenotypes observed after anti-SF injection were a spurious non-specific results, we injected embryos with an antibody against Dgt6 (a subunit of the Augmin complex involved in spindle formation) that had been prepared at the same time as the anti-SF antibodies. Consistent with previous results (Hayward et al., 2014), the embryos injected with the anti-Dgt6 antibodies showed delayed spindle formation, thinner and long spindles and arrested in a metaphase-like state. This phenotype is very different from that elicited by anti SF-antibodies, ruling out the possibility that the mitotic effects of the anti-SF antibodies are due to chemical contaminants associated with antibody purification and concentration processes.

[Editors' note: the author responses to the re-review follow.]

Summary:The manuscript by Gatti et al., addresses whether the splicing factors Sf3A2 and Prp31 have direct mitotic roles, independently from splicing. In this revised manuscript the authors have made considerable efforts to address previous criticisms of their study. It seems to me that they have done pretty much as much as they can to exclude the possibility that the splicing factors are acting indirectly – their analysis of transcripts is particularly important here. The biochemical data showing interaction between these two factors with MTs and Ndc80 offer a plausible explanation for some of the observed phenotypes and suggest that mitosis is immune to potential splicing defects in the several mRNAs that are under control of Sf3A2 and Prp31.Essential revisions:Before publication the authors should address several inconsistencies in the data.1) Ndc80 localization at kinetochores seems reduced after perturbation of Sf3A2 and Prp31only in the presence of MTs. Ndc80/Hec1 localization is normal without MTs (at least in HeLa cells). Thus, SFs are not required for Ndc80 recruitment but might be involved in its maintenance at KTs. Consistently, the in vivo interaction between Sf3A2 and Prp31 with Ndc80 does not seem to take place in mitotic cells in the absence of MTs, leading the authors to propose a model in which the Ndc80/Hec1-SF interaction requires the presence of spindle MTs. The problem comes from the GST pull-down experiments that clearly show that purified recombinant Ndc80 and SFs all interact, and this happens in the absence of MTs. Together with the fact that the observed mitotic phenotypes related with the perturbation of SFs (e.g. monopolar spindles) are not consistent with loss of Ndc80, leaves open an obvious gap in the model. This cannot simply be explained by differences in SAC activity among different cells/systems. These points should be addressed either by modifying the model to include caveats and alternative interpretations – notably that there may be other targets than Ndc80 – or by including experimental data to support the proposed model.

We agree with the reviewer that the fact that GST pulldowns with purified proteins show that Ndc80 directly interact with the SFs is somewhat at odds with the finding that human SFs interact with HEC1 only in the presence of MTs. However, the fact that two bacterially expressed, affinity purified proteins interact in vitro does not necessarily imply that they directly interact also in living cells. For example, it is possible that post-translational modifications occurring in living cells alter the binding abilities of these proteins. We have shown that the SFs interact with HEC1 only in the presence of spindle MTs and that in the absence of the SFs both *Drosophila* Ndc80 and human HEC1 tend to dissociate from the kinetochore and diffuse in the cytoplasm, possibly along the kinetochore MTs. We have therefore modified our model in the following way:

“Collectively, these results suggest a model for the mitotic role of SF3A2 and PRP31. We propose that these SFs interact with Ndc80/HEC1 on the k-fibers, regulating the association of the Ndc80 complex with the kinetochore. In the absence of the SFs, the Ndc80/HEC1 complex would associate with the kinetochores, or the kinetochore-MT interfaces, in an abnormal fashion, so that the binding of the complex to the kinetochore would be weakened and the SAC satisfaction prevented. As a result, when the kinetochores are attached to MTs, Ndc80/HEC1 would diffuse away from the kinetochores possibly along the kinetochore fibers.”

We also added that some aspects of SF-Ndc80/HEC1-MT interactions are reminiscent of those described for the mammalian Ska complex (see revised Discussion section).

Our revised model on the mitotic role of the SFs also explains why the phenotypes of SF depleted cells are different from those of cells lacking Ndc80/HEC1. In our model, Ndc80/HEC1 is associated with the kinetochores in an abnormal fashion, and this would result in a mitotic phenotype different from that caused by Ndc80 depletion.

2) The immunofluorescence data remains somewhat difficult to interpret. The authors state that detergent pre-extraction before fixation improves visualization of KT proteins, but this is also known to cause artefacts. The reviewers do not see the need for such harsh conditions and are aware that conventional PFA works well to stain Hec1; even methanol fixation is preferable to pre-extraction. The authors should include data using a different staining technique that avoids pre-extraction.

As suggested by the reviewers, we performed new experiments in which RNAi cells were fixed with PFA. As described in the Results section (see also new Figure 9) these experiments showed that HEC1 accumulation at the kinetochores was reduced, without a concomitant significant reduction in the amount of the protein (Figure 9C). We specifically observed that in most RNAi cells there was a halo of HEC1 in the cytoplasmic area surrounding the chromosomes (Figure 9D). Notably *SF3A2* and *PRP31* nocodazole-treated (for 3 hours) cells displayed HEC1 signals comparable to those of control cells and most of them did not show a fluorescent HEC1 halo around the chromosomes (Figure 9E). In RNAi cells fixed with PFA (and not treated with nocodazole), we did not observe an association of HEC1 with the spindle MTs like in cells pre-extracted with Triton X-100. In the revised text, we explained that the pattern of HEC1 localization observed in pre-extracted RNAi cells (but not in control cells) does not necessarily reflects a diffusion of HEC1 along the kinetochore MTs:

“However, although the most straightforward interpretation of our observations on Triton X pre-extracted cells suggests that HEC1 spreads out along the kinetochore MTs, we cannot exclude the possibility that the association of HEC1 with the spindle MTs depends on the fixation technique. It is indeed possible that HEC1 dissociates from the kinetochores and then binds the MTs that resisted the pre-extraction/fixation procedure.”